# The effect of 3D stereopsis and hand-tool alignment on learning effectiveness and skill transfer of a VR-based simulator for dental training

**Maximilian Kaluschke**[1], **Myat Su Yin**[2], **Peter Haddawy**[2]*, **Siriwan Suebnukarn**[3], **Gabriel Zachmann**[1]

**1** Computer Graphics and Virtual Reality, University of Bremen, Bremen, Germany, **2** Faculty of ICT, Mahidol University, Bangkok, Thailand, **3** Faculty of Dentistry, Thammasat University, Bangkok, Thailand

* peter.had@mahidol.ac.th

**Data Availability Statement:** The complete dataset, and R-script files are available from the FigShare database (URL https://figshare.com/s/

## Abstract

Recent years have seen the proliferation of VR-based dental simulators using a wide variety of different VR configurations with varying degrees of realism. Important aspects distinguishing VR hardware configurations are 3D stereoscopic rendering and visual alignment of the user's hands with the virtual tools. New dental simulators are often evaluated without analysing the impact of these simulation aspects. In this paper, we seek to determine the impact of 3D stereoscopic rendering and of hand-tool alignment on the teaching effectiveness and skill assessment accuracy of a VR dental simulator. We developed a bimanual simulator using an HMD and two haptic devices that provides an immersive environment with both 3D stereoscopic rendering and hand-tool alignment. We then independently controlled for each of the two aspects of the simulation. We trained four groups of students in root canal access opening using the simulator and measured the virtual and real learning gains. We quantified the real learning gains by pre- and post-testing using realistic plastic teeth and the virtual learning gains by scoring the training outcomes inside the simulator. We developed a scoring metric to automatically score the training outcomes that strongly correlates with experts' scoring of those outcomes. We found that hand-tool alignment has a positive impact on virtual and real learning gains, and improves the accuracy of skill assessment. We found that stereoscopic 3D had a negative impact on virtual and real learning gains, however it improves the accuracy of skill assessment. This finding is counter-intuitive, and we found eye-tooth distance to be a confounding variable of stereoscopic 3D, as it was significantly lower for the monoscopic 3D condition and negatively correlates with real learning gain. The results of our study provide valuable information for the future design of dental simulators, as well as simulators for other high-precision psycho-motor tasks.

## 1 Introduction

Development of expertise in dentistry requires extensive training of specific dexterous skills. A dental surgeon's skill increases with practice, as evidenced by the strong correlation between

6eddea3f66489716fba0, DOI number 10.6084/m9.figshare.22362322).

**Funding:** This work was partially supported by a grant from the Mahidol University Office of 638 International Relations to Haddawy in support of the Mahidol-Bremen Medical 639 Informatics Research Unit (MIRU), and by a fellowship from the Hanse-Wissenschaftskolleg 640 Institute for Advanced Study. No additional external funding was received for this study.

**Competing interests:** The authors have declared that no competing interests exist.

dental surgeon skill and practice time [1, 2]. Therefore, dental schools have long used different forms of simulation to provide students with practise opportunities for these particular skills before ever practicing on live patients. The early, and still most commonly used, simulators consist of a mannequin head (so-called phantom head) with plastic teeth. These simulators can be used by the students to practice various procedures. Depending on the procedure, the teeth are either simple inexpensive solid plastic teeth (for simple procedures such as caries removal) or more expensive plastic teeth with different layers and internal anatomy (for complex procedures, such as root canal access opening). Upon completion of a procedure, a dental instructor scores the outcome based on visual inspection. As the teeth are significantly altered during practice, they can only be used effectively for a single time, resulting in high operational cost.

In recent years, VR-based dental simulators have increased in popularity due to enabling technological advancements, combined with concrete benefits of the approach [3–5]. VR simulators offer high-fidelity simulations that are reusable, resulting in considerably lower operational costs, and they can be configured to provide trainees practice on a variety of different cases. They also have the ability to record accurate data on individual performance, which provides the opportunity for trainees to receive objective feedback to facilitate learning. VR simulators show significant real-world learning effects for the virtually trained surgical procedures [6–8]. In addition, medical trainers' growing need for objective and automated assessment tools [9, 10] could be addressed by VR-based simulators. In contrast to plastic teeth, simulator outcomes can be scored automatically [11], provided the simulator is suitably designed and implemented.

The requirements for a VR simulator to be an effective teaching tool and to be an effective assessment tool are closely related but distinct. To be an effective teaching tool, practice time in the simulator must translate into significant improvement in real-world performance. To be an effective assessment tool, real-world skill level must translate into simulator performance, without requiring significant time to learn the idiosyncrasies of the simulator. It is possible for a simulator to satisfy one of the requirements but not the other. For example, a simulator that is difficult to use may still result in real-world performance gains, yet not be useful as an assessment tool. This was the case in some early VR dental simulators that displayed results on a 2D monitor [12]. These two requirements can be thought of in terms of two types of transferability: simulator to the real-world and real-world to the simulator.

With the variety of VR technologies available, dental simulators have been developed using a wide variety of different VR configurations. Display technologies used include traditional 2D monitors, 3D monitors, half-mirrored displays, and head-mounted displays (HMD), the latter three of which provide stereoscopic depth perception. Instrument manipulation is achieved with and without haptic feedback. In addition, the use of HMDs and half-mirrored displays supports hand-tool alignment in which the user sees the dental instrument in the same location as their physical hand. In contrast, 2D and 3D monitors do not provide such alignment. While each new dental simulator typically is associated with some form of evaluation study [13–15], only few comparative studies have been carried out to determine the benefits of the simulation aspects associated with the various available VR technologies being used and none examine the impact of those factors on teaching effectiveness or assessment suitability, as measured by transferability.

In this paper we examine the impact of the two major selling features of HMDs for virtual simulators, 3D stereoscopic rendering and hand-tool alignment, on the teaching effectiveness and the suitability as an assessment tool of a VR dental simulator. These features are not possible to achieve with traditional monitors or 3D monitors. Students were trained on an immersive VR simulator while systematically and independently controlling for each of the two

aspects of the simulation. Learning gains were measured in two ways, by doing pre- and post-testing on realistic plastic teeth, as well as by assessing the virtual training outcomes using a novel automated scoring metric.

We measure teaching effectiveness in terms of learning gains between pre- and post-testing on realistic plastic teeth. We measure suitability for assessment, in terms of the correlation between real-world pre-testing score and the virtual score of the first simulator session following the pre-testing. Based on these metrics and the two VR technology aspects (stereoscopic 3D vision and hand-tool alignment), we formulate four hypotheses:

$H_{V_{learn}}$   Stereoscopic vision has a positive impact on the learning effectiveness of the simulator, as measured by real learning gains.

$H_{A_{learn}}$   Hand-tool alignment has a positive impact on the learning effectiveness of the simulator, as measured by real learning gains.

$H_{V_{assess}}$   Stereoscopic vision has a positive impact on the simulator's suitability for assessment, as measured by initial simulator performance.

$H_{A_{assess}}$   Hand-tool alignment has a positive impact on the simulator's suitability for assessment, as measured by initial simulator performance.

## 2 Related work

With an increasing trend of using 3D stereo-projected images to create realistic virtual learning environments, there is an ongoing debate as to whether stereo-projected images are a necessary feature of simulators [16–18]. A comprehensive review conducted by McIntire et al. [16], found that in 15% of over 180 experiments from 160 publications, stereoscopic 3D display either showed a marginal benefit over a 2D display or the results were mixed or unclear, while in 25% of experiments, stereoscopic 3D display showed no benefit over non-stereo 2D viewing. They concluded that stereoscopic 3D displays are most useful for tasks involving the manipulation of objects and for finding/identifying/classifying objects or imagery. The majority of these studies used 3D monitors for the stereoscopic 3D condition and displayed the same image to both eyes or used 2D monitors for the monoscopic 3D condition. Buckthought et al. [19] showed that dynamic perspective changes enhance depth ordering performance. Therefore, the depth information conveyed through monoscopic 3D inside an HMD which can be freely moved and moved closer and further could provide more helpful depth information when compared to 2D monitors, as the dynamic perspective changes provide depth cues.

de Boer et al. [20] investigate the differences in students' performance in carrying out manual dexterity exercises with the Simodont dental trainer simulator (The MOOG Industrial Group; www.moog.com) in 2D and 3D versions. 3D vision in the dental trainer was based on the projection of two images superimposed onto the same screen through a polarising filter. 2D vision was obtained by turning off one of the two projectors such that only one image was projected onto the screen. All of the students in both the 2D and 3D vision groups wore polarised glasses during the practice sessions and when testing to keep the environmental factors constant. The task consisted of using a dental drill to remove material from a cube and the outcomes were automatically scored. The results showed that students working with 3D vision achieved significantly better results than students who worked in 2D. In an administered questionnaire, participants also indicated that they preferred the 3D vision setting. Students reported having an unpleasant experience in working with 2D vision while wearing the glasses. The probable reason is that only one eye received an image through the polarized glasses. In a

related study, Al-Saud and colleagues [21] examined the effects of stereopsis on dentists' performance with the Simodont dental simulator. Thirteen qualified dentists were recruited and asked to performed a total of four different dental manual dexterity tasks under non-stereoscopic and stereoscopic vision conditions with direct and indirect (mirror) observation. The tasks consisted of removal of material from a geometric shape embedded in a cube of material. Automated scoring was based on amounts of target and non-target material removed. Stereoscopic 3D was the simulator's normal operation and was achieved as in the previously mentioned study [20]. To produce 2D images, the simulator was engineered to output a single image to both eyes. The study found out that depth related errors were significantly higher under non-stereoscopic viewing but lateral errors did not differ between conditions. Both studies used the commercial Simodont simulator on a 3D monitor (which displays monoscopic 3D in one condition, thereby acting like a 2D monitor). 3D monitors do not allow for unrestricted head tracking and do not support hand-tool alignment, which our simulator supports through the use of HMD and calibration of the haptic devices.

Collaco et al. [22] investigated the effects of (full) immersion and haptic feedback on inferior alveolar nerve anesthesia technical skills training. Their experimental study consists of preceptorship and training phases. During the preceptorship phase, one of the groups received the anesthesia instructions from the dental instructor on a full HD TV screen, while the participants from the remaining three groups observed the anesthesia procedure from the instructor's perspective in an immersive condition using the HMD. In the training phase, the participants in one of the groups in the immersive condition during the previous preceptorship stage performed the anesthesia injection using the full HD TV screen while the remaining three groups performed the task with the HMD in the immersive condition. The results showed that participants without immersive displays had less accurate needle insertion points, though needle injection angle and depth were not significantly different between the groups. The needle insertion point here needs to be found without haptic feedback. As such it differs considerably from the root canal opening, since the bur can touch the tooth with drilling disabled for orientation with the help of haptic feedback. Due to these differences we expect stereo vision to have a smaller positive effect on performance and on learning.

In manipulating tools, users receive information from two feedback loops: the body-related proximal feedback loop (proximal action effect) such as tactile sensations from the moving hand, and from the effect in distal space, such as the visual feedback from the movement of effect points of the tool (distal action effect). Establishing the mapping between the moving hand and the moving effect part of the tool can add challenges to the human information processing systems. According to Sutter et al. [23], if information from proximal and distal feedback loops are equally important for controlling actions, any discrepancy between them would be a constant source of interference to the user. Users of conventional desktop-based VR simulators using haptic interfaces are familiar with this scenario while manipulating the haptic device and observing the action effects on a display monitor. Meanwhile, in HMD simulators, the spatial gap between the hand and the resulting movement can be eliminated by manipulating the virtual camera position and rotation to the user in such a way that the user sees and feels as if he is manipulating the dental tools on the patient's teeth. Although more realistic, it is interesting to note that in this condition the vision may be afforded with a higher weighting than other sensory information; a situation often referred to as visual capture. Although visual information is invaluable for executing skillful manual tasks, visual capture can produce powerful illusory effects with individuals misjudging the size and position of their hands. Moreover, if vision of the hand/tool is available in the operating area it should be recognized that there might well be interference that would impair motor performance and learning, as there is a shift in attentional focus to the outcome of actions rather than the actions themselves.

Wilkie et al. [24] studied whether visual capture can interfere with an individual's rate of motor learning in a laparoscopic surgery setting. They investigated the adaptation to distorted visual feedback in two groups: a direct group directly viewed the input device, while an indirect group used the same input device but viewed their movements on a remote screen. When distortion exists between hand and tool movement, then visual capture is an issue and participants in the indirect group performed better than those in the direct group. However, when no distortions were applied, participants in the direct group performed better than participants in the indirect group. In the dental domain, there is typically no distortion present for drilling tasks. Similarly, Sutter et al. [23] conducted several experiments aiming to investigate the underlying motor and cognitive processes and the limitations of visual predominance in tool actions. Their major finding is that when transformations are in effect the awareness of one's own actions is quite low. These findings suggest that hand-tool alignment will have a profound effect in our user study on learning effect and performance.

The effect of stereoscopic vision inside an HMD on dental surgery simulator suitability for assessment, user performance, and skill transfer have not been investigated previously. Even in the context of arbitrary use-cases, stereoscopic 3D inside HMDs has not been investigated systematically by using the same technology but removing the depth cue of stereopsis. Additionally, the effects of hand-tool alignment have also not been investigated yet, although it is a prominent feature in modern dental surgery simulators. This study attempts to fill both of these gaps.

## 3 Simulator

We developed a VR dental surgery simulator with haptic feedback, in which students can practice caries removal, crown preparation, and root canal access opening (see Fig 1, for the students' perspective see Fig 2). The simulator was developed using Unreal Engine (UE) 4.26. An HTC Vive Pro Eye with a combined resolution of $2880 \times 1600$ and eye sensors was used to display stereo images from the UE SteamVR plugin. Eye tracking of the HMD user was implemented using the SRanipal Unreal plugin. The dental virtual hand-piece and mirror are each controlled by a Geo-Magic Touch haptic device (Phantom) with 6 degrees-of-freedom (DOF) input and 3 DOF output. Haptic feedback is provided to simulate the interaction between the hand piece and virtual tooth. The sound of the drill is also simulated. The virtual patient was modeled using the Metahuman framework [25] and imported into our UE scene. The virtual human is rendered with high fidelity visuals including subtle idle animations of the face and mouth, such as eye blinking and movement of the tongue. We made sure to not include animations that would alter the location of the tooth. We added a transparency texture to the virtual teeth texture, which allows us to hide one of the teeth (#36) of the Metahuman model. In its place, we inserted a new tooth that we modeled by hand with guidance from CT scans of similar teeth and approved by an expert dentist. At runtime, we render the tooth by using the UE Procedural Mesh Component (PMC). We generate triangles of modified tooth regions in a CUDA library, which are then fed to UE's PMC. The library approximates the tooth surface by a metaball surface that is discretized at runtime using a parallel marching cubes implementation with a resolution of $90 \times 135 \times 90$. We compute the haptic feedback outside of the UE main loop, so as not to be limited by the rendering frame rate. The force is computed according to the algorithm presented in [26], which uses an inner spheres volume representation. The tooth enamel is made up of 100k, the dentin by 170k, and the pulp by 10k spheres. We tuned the force, drilling, and friction parameters by our subjective impression of drilling the real plastic teeth that students usually practice on, with approval by an expert dentist. These plastic teeth closely resemble the feeling of drilling real teeth and are anatomically correct.

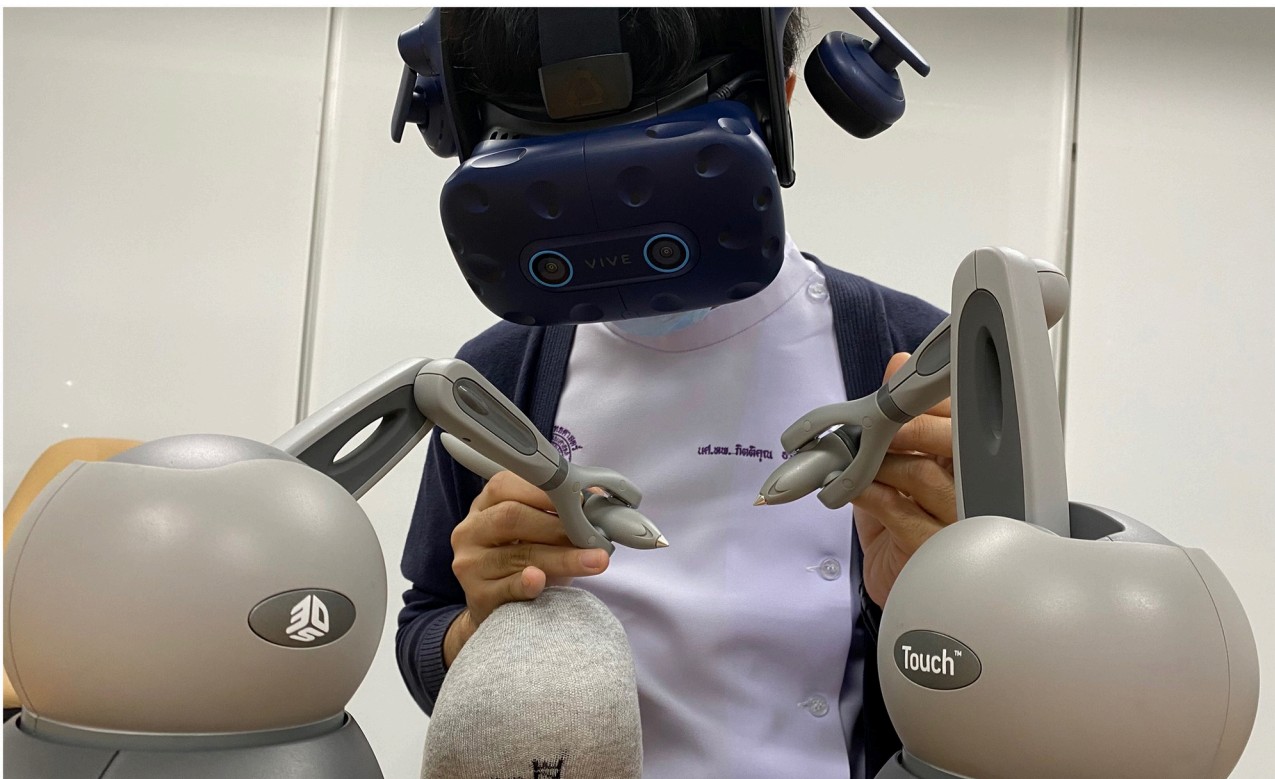

**Fig 1. The VR dental surgery simulator is used by a dentistry student to practice root canal access opening on tooth #36.** The VR HMD and haptic input/output devices allow for a intuitive control with realistic haptic feedback (in alignment condition). The monitor shows the image that the student is seeing on the HMD.

### 3.1 Stereo rendering

The standard VR rendering is set up to be at a realistic scale, such that the user has a natural stereo impression from the two different images that are sent to the eye. This setting will later be referenced as the "stereo" condition during the user study. To investigate the effect that stereo vision has on the learning effect, we implemented a rendering mode that renders the virtual scene without stereoscopy. We implemented monoscopic 3D by rendering identical images for the left and right eye. This setting will later be referenced as the "mono" condition during the user study. Another possibility to achieve monoscopic 3D is to have a screen-space shader that blanks out one eye. However, we found, similarly to [20], that it creates an unpleasant feeling.

### 3.2 Hand-tool alignment

The force feedback devices are registered with the HTC Vive VR system by using a VR controller dock that is mounted on a board with a static offset to both haptic device bases (Fig 3 shows the misalignment condition). Inside the game engine, we define the virtual position of the haptic device origins of the mirror ($p_M$) and the drill ($p_D$) inside the scene, with the virtual distance set to the physical distance between them, 30 cm in our setup. When we run the simulator in a new VR configuration (new light house locations or new haptic device locations), a calibration procedure is manually initiated by a key-press. We calculate the virtual VR controller target origin $p_{C_T}$ by applying an offset to the mirror origin, that we previously defined. The offset

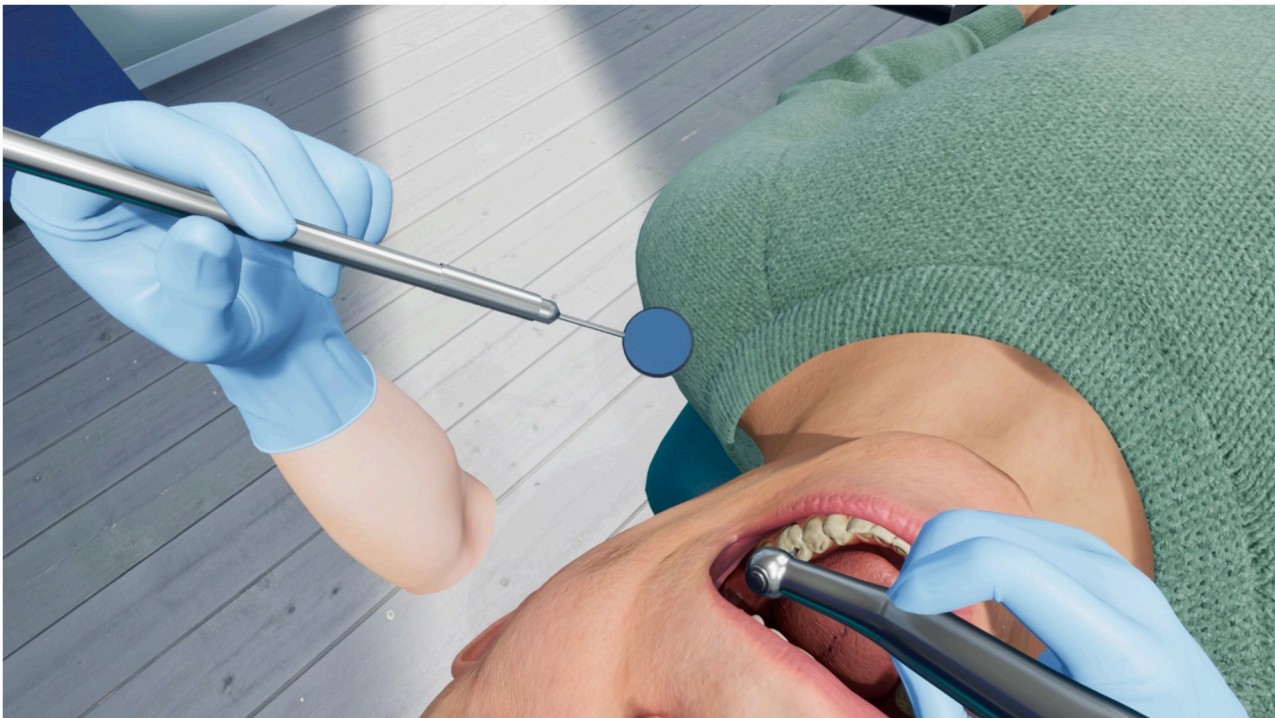

**Fig 2. In-game view of the VR dental surgery simulator, in which a student is performing root canal access opening on tooth #36.**

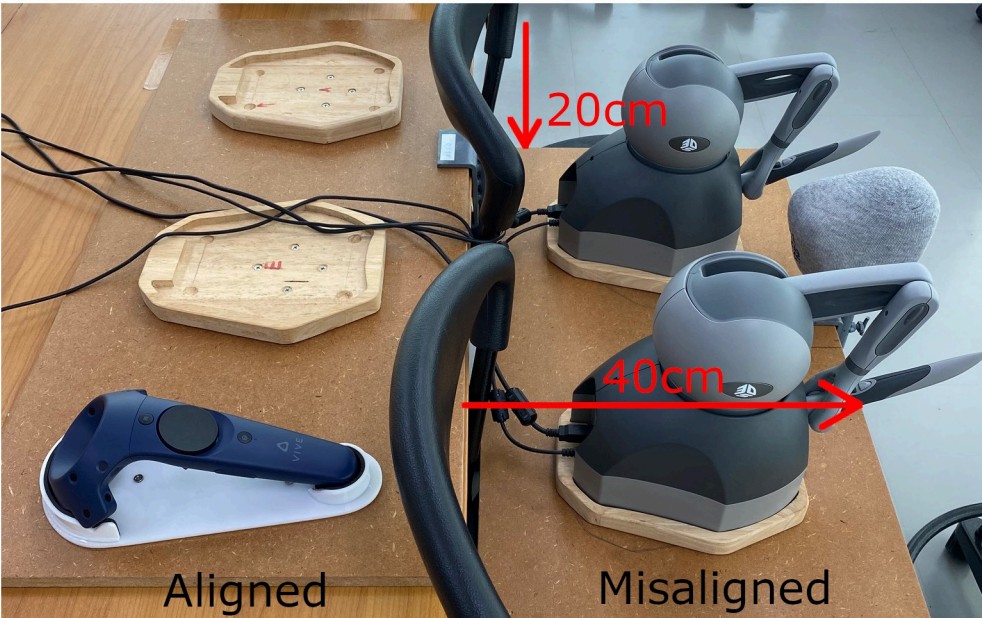

**Fig 3. The calibration of the haptic devices with the HTC Vive VR system is implemented using a VR controller with a static offset.** The "hand-tool misalignment" is achieved by calibrating and then moving the haptic devices forward and downward in front of the table, as shown here. Republished from [27] under a CC BY license, with permission from IEEE, original copyright 2022.

needs to be measured and tuned by hand, in our setup, the offset is a translation of $\Delta p = (22cm, 26cm, -7cm)$ and a rotation of $\Delta\theta = (0°, 0°, 90°)$. We then define the target VR controller origin as

$$p_{C_T} = p_M + \Delta p \tag{1}$$

$$\theta_{C_T} = \theta_M + \Delta\theta \tag{2}$$

where the rotation angles are simply added together. Now given the virtual target VR controller location $p_{C_T}$ and actual physical VR controller location $p_C$, we calculate the difference and add it the VR camera location $p_{VR}$:

$$p'_{VR} = p_{C_T} - p_C \tag{3}$$

$$\theta'_{VR} = \text{Delta}(\theta_{C_T}, \theta_C) \tag{4}$$

where Delta(A,B) calculates the difference by subtraction $A - B$ followed by normalization to the range of $[-180, 180]$. By doing this, we align the virtual tools and haptic device handles, within the accuracy of the VR tracking. We call this condition "hand-tool alignment" (as shown in Fig 1).

To define the contrasting condition, "hand-tool misalignment", we do the same calibration, but additionally offset the real haptic devices. We moved the haptic devices down by 20 cm and forward by 50 cm, relative to the table (see Fig 3). We chose this offset to simulate a misalignment setting that resembles the offset on a desktop monitor in VR.

## 3.3 Visual perception

Dentists make heavy use of their eyes, during dental surgery, such as to check bur depth and bur orientation, as well as in pauses that occur between drilling a tooth, to precisely inspect their own progress. Modern HMDs allow for easy tracking of gaze behavior, with appropriate sensors already built-in, which is the case for the HTC Vive Pro Eye. This made eye tracking easy to implement into our simulator, however the accuracy was a challenge. Since the objective of this study required only determining the the eye-tooth distance, we used a simple form of eye tracking to determine at which point in time the user is looking at the tooth and where his eyes are located. Based on the "cyclops eye" (it is the mid-way point between the left and right eye, here in world coordinates) and tooth position, we can determine if the gaze ray hits the tooth, and in those instances, we log the current eye position and tooth hit position. Using this data, we determined the mean eye-tooth distance over an entire trial and regarded each trial as a separate sample point.

The human eye can naturally see much more detail than the HTC Vive Pro Eye can display with its limited resolution. This is very apparent when looking at small objects in VR, such as a human tooth and its individual features such as the root canal orifices. Since the accommodation range puts a lower bound on the distance of our eyes to the tooth, the screen resolution of the tooth is highly limited. If one looks at Fig 2, one can see that at a viewing distance of around 23 cm, the tooth takes up a miniscule amount of the screen. We estimate the area to be around $119 \times 119$ pixels, taking up only 0.31% of the already limited HMD screen resolution. For healthy people between 20 and 25 years old, the accommodation near point and convergence near point are 9.92 cm and 7.18 cm [28], which set a physical limit for how closely objects can be focused. However, in case of stereo vision, we suspect that this lower bound is much higher inside a VR HMD, such as the HTC Vive Pro Eye. From our subjective tests, the

near point that can be focused on is in the range of 20 to 25 cm. One possible explanation for this could be that the HMD's limited field of view increases the stereo disparity and makes interocular correlation especially difficult, which limits the range in which binocular vision works effectively [29]. For the monoscopic 3D condition, the stereo disparity is always 0, no matter how close or far objects are. Therefore, there is no lower bound for the distance that objects can be focused on in monoscopic 3D, so participants of this condition can move as close as they desire to the tooth, unlike participants within the stereoscopic 3D condition. Based on the lower focus bound in the stereo 3D condition, we expect the eye-tooth distance to be lower for stereo 3D, with an average distance around 20 to 25 cm.

## 3.4 Automated outcome scoring

Dental outcomes are usually scored by an expert in dentistry. This score might appear subjective, however they follow a close set of objective measures, which makes it a robust scoring system that is mostly objective. For example, when we let two independent expert dentists score our data set, the experts' scores had excellent reliability ($\kappa = 0.87$, and intra-class correlation (ICC) of 0.98). In the dental scoring system, each of the four cardinal tooth walls and the pulp floor is visually observed and rated for errors by the expert. The criteria for rating the errors can be summed up in the following way:

+0 Access to all orifices without an excess cavity.

+1 Access to all orifices with minor over-drilling.

+2 Incomplete removal of pulp chamber roof and/or excessive over-drilling.

+3 Unidentified canals and/ or perforation.

The overall error rating for a tooth is taken to be the sum of the error ratings of the walls and pulp floor. Therefore, the error ranges from 0 to 15, with lower values indicating better performance (examples shown in Fig 4). Based on the excellent conformity of the two experts, we used the mean error value in our analysis. It would augment the simulator to implement an automated scoring system based on the outcomes achieved inside the simulator. Our automated score should highly correlate with the experts rating of those virtual outcomes. However, as our user study is comprised of 40 participants, each running 6 trials, we have 240 total outcomes. It was not feasible to ask the experts to evaluate each one of the 240 virtual outcomes, as it is too much data. Therefore, we needed to compress the data set to essential

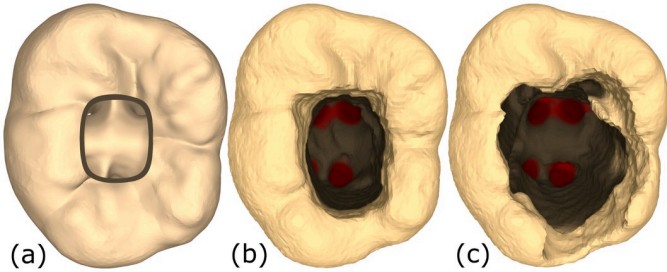

**Fig 4. Different conditions of tooth #36.** (a) Ideal root-canal access opening. (b) A root canal access opening with a low error score. All orifices are accessible with little over-drilling. (c) A root canal access opening with a high error score, as multiple walls are over-drilled and not smooth. Republished from [27] under a CC BY license, with permission from IEEE, original copyright 2022.

outcomes that sample the complete value range that all of the 240 outcomes encompass as uniformly as possible. We proceeded with the implementation in four steps:

i).  Generate an ideal drilling outcome. We generated the tooth in Fig 4 (left) by consulting an expert dentist, to verify that there is no under-drilling or over-drilling present, and that all walls and floor are well-shaped and have smooth edges. All four orifices are visible from the access opening (though not necessarily from the same angle).

ii).  Select from existing binary classification metrics one that generates normally distributed scores for the total outcome range. Additionally, we manually checked random samples visually to check if they are sensible based on the previously shown expert scoring system, evaluated by a dental novice. Here we looked at 24 of the state-of-the-art metrics (many of which are presented in [30]) and selected the F1-score [31] to be most ideal for further processing.

iii).  Select 20 samples that uniformly cover a wide range of the total value range with the previously chosen metric (F1-score) and let experts score these outcomes, without knowledge of the F1-score. The experts received each sample as a 3D mesh, which they could rotate and inspect on their personal computer. Again, the experts had excellent reliability ($\kappa = 0.89$, and ICC = 0.998).

iiii).  Implement a new metric and fine-tune it such that correlates well with the expert scores.

Through exploration we found that the F1-score, which is the harmonic mean of Sensitivity and Precision, can be improved upon. We developed a new metric we call *Dentist* (abbreviated by *D*), which combines the two scores of Sensitivity *S* and Precision *P*

$$P = \frac{TP}{TP + FP} \quad , \quad S = \frac{TP}{TP + FN}$$

we adjust the value range by linear interpolation

$$\tilde{P} = \frac{P - 0.95}{1 - 0.95} \quad , \quad \tilde{S} = \frac{S - 0.2}{1 - 0.2}$$

given those, we define Dentist *D* as

$$D = \left(1 - \frac{1.5\tilde{S} + \tilde{P}}{2.5}\right)15 \tag{5}$$

$$= \frac{15\,(32 \cdot FP \cdot TP + 3 \cdot FN \cdot TP + 35 \cdot FN \cdot FP)}{4\,(TP + FN)(TP + FP)} \tag{6}$$

It is essentially a weighted mean of $\tilde{S}$ and $\tilde{P}$, though the values are flipped to represent a distance rather than a similarity, as well as multiplied by 15 to match the dentists' rating system. The value ranges of *S* and *P* are adjusted, because $P \in (0.96, 0.995]$ whereas $S \in (0.2, 1)$. Therefore we adjusted both components to occupy roughly the same value range, the full range [0, 1].

We found *S* correlates well with penalization of over-drilling, whereas *P* correlates well with penalization of under-drilling. The two metrics complement each other well, as can visually be inspected in the individual metric correlation plot in Fig 5. When looking at the individual correlations, the highs and lows of both functions balance out to be nearly straight in our weighted sum. Interestingly, most metrics exhibit a similar shape to that of the Sensitivity

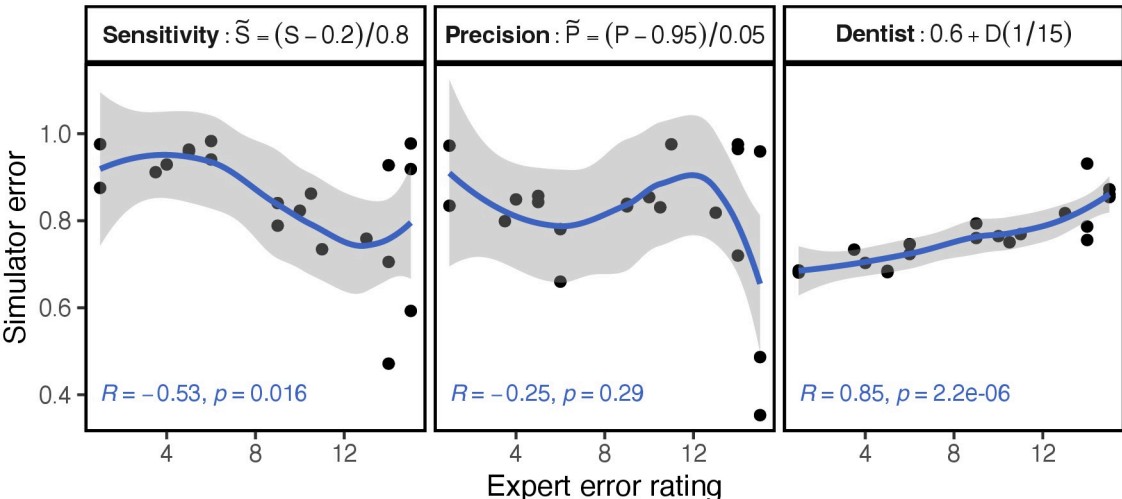

**Fig 5. Correlation of our metric and the basic metrics.** The line is a polynomial regression to illustrate the different curvatures. There is a high correlation of our scoring metric with the ratings of two independent experts'. In this graph, we show the metric in the value range of [0, 1], with 1 as ideal, to compare with other similarity metrics. In general however, the scoring metric is chosen to be in the range of [0, 15], with 0 as the ideal outcome, same as the dentist' rating system.

correlation, so most would probably penalize over-drilling more. At runtime, we extract 3D voxels from the inner spheres volume by defining an implicit surface and discretizing it on a $90 \times 135 \times 90$ grid. The same data is also used to generate the triangles, normals and colors to represent the rendered geometry at runtime. The extraction needed spatio-temporal optimization to run at interactive rates. Based on these voxel values, we can compute the standard binary voxel classification sums:

- *TP* (True Positive): Correctly undrilled voxels.

- *TN* (True Negative): Correctly drilled voxels.

- *FP* (False Positive): Incorrectly undrilled voxels.

- *FN* (False Negative): Incorrectly drilled voxels.

As we can see from their respective definitions, *FN*, which is the penalty count for over-drilling, is only found in *S*, which explains why it penalizes over-drilling. *P* penalizes under-drilling more as it only incorporates *FP* as the measure for error. The correlation of *D* with the expert rating is of high degree with $R = 0.85$, $p < 0.0001$ (see Fig 5). Of the 24 common classification metrics that we evaluated, the best reach a correlation of approx. $-0.65$. Another existing scoring method achieves an information-based measure of disagreement (IBMD) [32] of 0.04-0.21 [11]. Besides an ideal drilling outcome, an expert has to additionally contract and expand the drilling region to create a min and max region that is used as weights in the nonlinear scoring function. Contrary to that, our method only requires a single ideal drilling outcome to compare against and achieves a similarly low IBMD of 0.09 (for the 20 essential outcomes measured against the expert ground truth). We did not use the IBMD at the scoring design stage as it measures absolute error, but for our user study, we find relative correctness to be more important. In the following the training score as well as the training gain will be calculated based on the Dentist metric. A future improvement could be to acquire more rated samples and use supervised learning to better approximate the experts' rating system.

## 4 User study

After receiving ethical approval from the Institutional Review Board from Mahidol and Thammasat universities, we invited students enrolled in the Faculty of Dentistry of Thammasat University to participate in our study. We recruited 40 participants (12 male, 28 female) and conducted a randomized controlled study. All participants were fifth year dental students, between 20 and 24 years of age and gave verbal consent to record anonymized data. They were not admitted to the study if any of the following criteria were present: (i) had received prior experience with the simulation, or (ii) received below 70% marks in knowledge assessment of endodontic cavity preparation, as this indicates insufficient theoretical knowledge to start practicing motor skill. The participants were randomly assigned to one of the four groups:

- Group 1: Stereoscopic 3D & hand-tool alignment

- Group 2: Monoscopic 3D & hand-tool alignment

- Group 3: Stereoscopic 3D & hand-tool misalignment

- Group 4: Monoscopic 3D & hand-tool misalignment

The task for the participants was to perform access opening on the virtual tooth during the training session and on a plastic tooth (lower left molar; tooth number 36; http://www.nissin-dental.net/) in pre- and post-training assessment sessions. A student's ability to perform the root canal access opening on such plastic teeth will predict with high reliability their ability to perform the task on real human teeth. Participants were briefly instructed on the use of the simulator, the experiment flow and the requirements of the access opening. As shown in the study flowchart (Fig 6), the training of each participant took place on two separate days. The

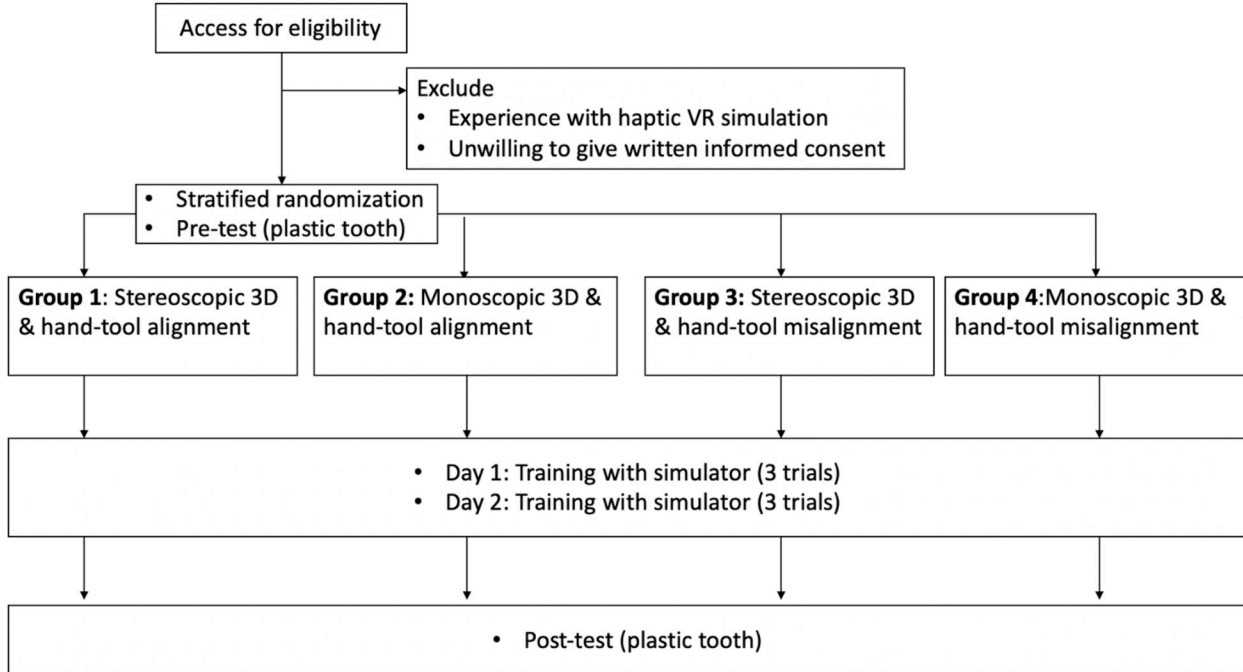

**Fig 6. Flowchart that shows the user study procedure.** Republished from [27] under a CC BY license, with permission from IEEE, original copyright 2022.

first day consisted of briefing, pre-test, and the first training session consisting of three trials using the simulator. The time for each trial inside the simulator on the first day was an average of 7.71 min (ranging from 1.11 to 26.53 min). After each trial, students could inspect their drilling result in detail on a separate computer screen (see Fig 4), which is not included in the above times. The second training session of three trials with the simulator, along with the follow-up post-test and answering two questionaires, took place afterwards on day 2. Here, the trials took an average of 5.18 min (ranging from 1.79 to 15.51 min), which is significantly faster than on the first day ($t(178.15) = 3.8$, $p < 0.001$). There was a gap of four to seven days between days 1 and 2 of training. The pre- and post-test plastic teeth were independently scored by two experts. As we mentioned in section 3.4, the individual scores had overwhelming conformity. Therefore, we used the mean value of the two experts' scores in the following analysis.

## 5 Results

The error scores for the pre-test range from 1 to 6.5, whereas the post-test scores range from 0 to 7 (see Fig 7). We define the error change $e_\Delta$ for each student as the difference between pre-test error score, $e_0$, and post-test error score, $e_1$, so $e_\Delta = e_1 - e_0$. With this, $e_\Delta$ defines the inverse learning gain for each student. The learning gain is normally distributed around the mean $M = -0.375$ with a median of $-0.5$ and standard deviation $SD = 1.87$. The value range is $-5$ to 4. We determined 3 outliers based on inter-quartile range analysis, resulting in removing the following learning gains: $\{-5, -5, +4\}$. These outliers are very unusually high and low learning gains which we feel do not represent an effect of the participant group but rather an inherent property of the participant. After removing outliers, the distribution is centered around the slightly larger $M = -0.24$ with the same median of $-0.5$ and $SD = 1.43$.

Looking at the pre- and post-error, we observe an overall decrease of students' error score from pre- ($M = 2.77$, $SD = 1.19$) to post-training ($M = 2.53$, $SD = 1.56$) of the root canal access opening inside our simulator. A paired one-tailed t-test shows a mean difference of $-0.2432$, with significance of $p = 0.153$ ($t(36) = 1.037$). Based on the $p$-value, we can not determine whether the students' overall improvement in performance is caused by the training. On the

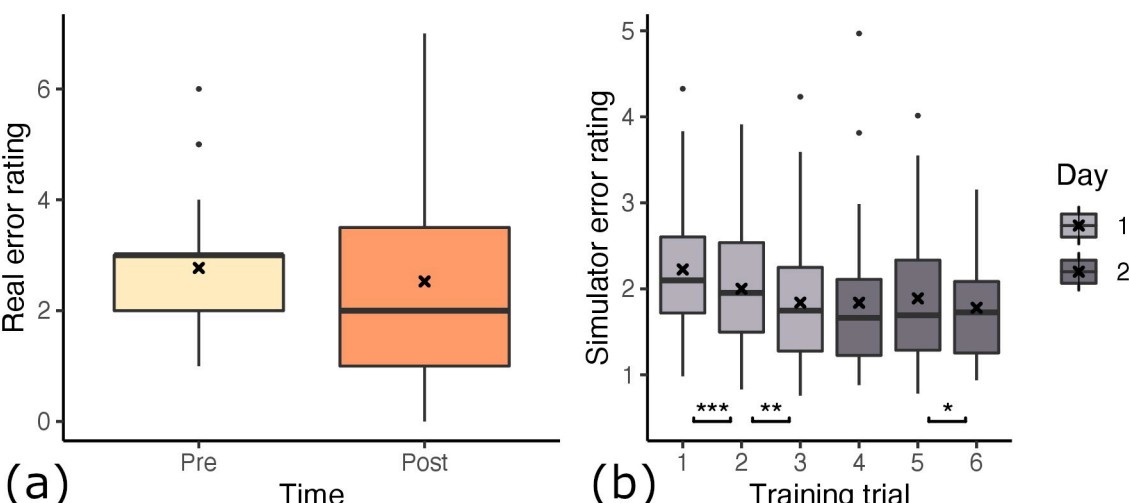

**Fig 7. Differences in paired error ratings with respect to time.** Gathered over all participants, regardless of condition groups. There was a 4-7 day wash out period between day 1 and 2. (a) Error determined by expert dentists on real outcome. (b) Error determined by algorithm on simulator outcome. Consecutive trials are compared for differences in means (*: $p < 0.05$, **: $p < 0.01$, ***: $p < 0.001$).

other hand, the participants' scores, measured inside the simulator (see Section 3.4 for scoring details), improved on average. However, here the score was significantly better at trial 6 compared to trial 1 ($t(36) = 14.7$, $p < 0.0001$). In fact, when looking at each trial score individually (see Fig 7b), we see a significant improvement from trial 1 to 2 of −0.22 ($t(36) = −3.38$, $p < 0.001$), from trial 2 to 3 of −0.16 ($t(36) = −2.85$, $p < 0.01$) and from trial 5 to 6 of −0.11 ($t(36) = −1.77$, $p < 0.05$). From trial #3 to #4 and from #4 to #5, we observed no improvement in the simulator outcome scores.

## 5.1 Groups

Between the four groups (as detailed in 4) we found differences in how well participants learned the task of root canal access opening. To determine the learning effect we compare each participants' pre-test error score to their post-test error score. The statistical significance is determined here by a paired one-tailed t-test with the hypothesis that the post-test error scores are lower than the paired pre-test error scores. As the learning gain is normally distributed, we used the parametric t-test. The distribution of pre- and post-test error rating per group are visualized in Fig 8a. The significant tests showed that none of the learning effects of the four groups are statistically significant.

We found that participants of group 1 "stereo & aligned" performed slightly better at the post-test ($M = 2.33$, $SD = 0.90$) compared to the pre-test ($M = 2.72$, $SD = 0.97$) with a mean difference of −0.389 ($t(8) = 0.902$, $p = 0.197$). Participants of group 2 "mono & aligned" improved their drilling performance between pre- ($M = 2.7$, $SD = 1.25$) and post-test ($M = 2.1$, $SD = 1.45$). The difference in error score of −0.6 is substantial ($t(9) = 1.327$, $p = 0.109$). Participants of group 3 "stereo & misaligned" on average scored worse in the post-test ($M = 3.5$, $SD = 2.12$) compared to the pre-test ($M = 3.2$, $SD = 1.57$) with a mean difference of 0.3 ($t(9) = 0.586$, $p = 0.714$). The scores of participants in group 4 "mono & misalignment" improved in the post-test ($M = 2.06$, $SD = 1.08$) compared to the pre-test ($M = 2.38$, $SD = 0.744$). This is an improvement of −0.313 in the error score ($t(7) = 0.637$, $p = 0.272$). A one-way ANOVA showed no statistically significant differences between the mean learning gains of the groups ($F(1, 35) = 0.335$, $p = 0.555$).

Interestingly, the simulator score changes showed different results (see Fig 8b). Here, all groups except group 1 increased their simulator scores significantly between the first (#1) and last trial (#6). The group with stereoscopic 3D and hand-tool alignment (group 1) did not improve or worsen their score significantly, going from $M = 1.79$, $SD = 0.52$ to $M = 2.20$, $SD = 0.7$ ($t(8) = −2.15$, $p = 0.968$). When doing a non-paired t-test between the simulator score

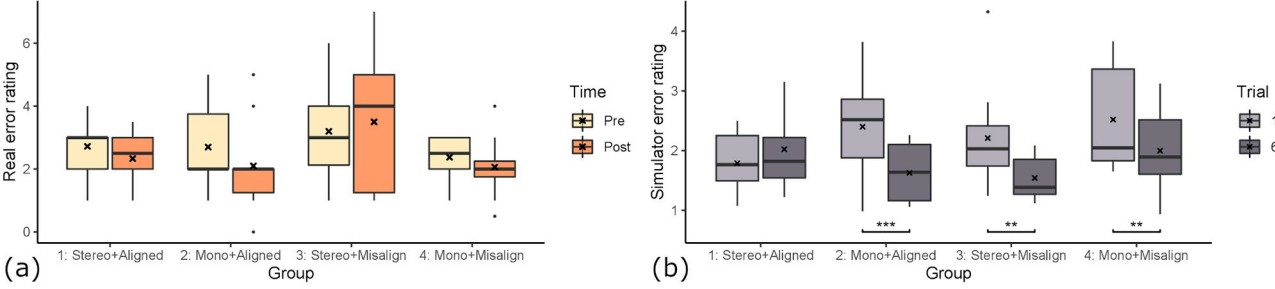

**Fig 8. Group influence on learning gain.** The improvement of the groups after 6 training trials inside the simulator. (a) Error determined by expert dentists on real outcome. Groups 1 & 3 improved less than 2 & 4. (b) Error determined by algorithm on simulator outcome. Groups 2,3 & 4 improved significantly (**: $p < 0.01$, ***: $p < 0.001$).

of group 1 on trial #1 and any groups simulator score at trial #6, there is no significant difference. The group with monoscopic 3D and hand-tool alignment (group 2) improved significantly from $M = 2.4$, $SD = 0.84$ to $M = 1.63$, $SD = 0.49$ ($t(9) = 4.78$, $p < 0.001$). The group with stereoscopic 3D and hand-tool misalignment (group 3) improved significantly from $M = 2.21$, $SD = 0.88$ to $M = 1.54$, $SD = 0.36$ ($t(9) = 3.44$, $p < 0.01$). The group with monoscopic 3D and hand-tool misalignment (group 4) improved significantly from $M = 2.52$, $SD = 0.91$ to $M = 2.0$, $SD = 0.73$ ($t(7) = 4.05$, $p < 0.01$). The fact that group 1 is the only group that did not improve their simulator score could indicate that the group 1 setting (stereoscopic 3D and hand-tool alignment) is the easiest to learn, as their trial #6 simulator scores do not significantly differ from the other groups' score. We explore this thought more in section 5.4.

When doing a one-way ANOVA of the learning gains, measured with simulator error rating, there is no significant difference between the groups ($F(1, 35) = 2.42$, $p = 0.129$). Therefore, there are no statistically significant differences between the mean learning gains when measured inside the simulator.

## 5.2 3D rendering modes

To examine the effect that stereoscopic rendering had on the participants' performance (see Fig 9), we regard the data of group 1 & 3 as one set of data ("stereo"), and 2 & 4 as the other set of data ("mono"). We thereby control for the alignment condition. The "stereo" group's pre-test error ratings ($M = 2.97$, $SD = 1.31$) decreased by 0.0263 for the post-test ($M = 2.95$, $SD = 1.72$). The one-tailed t-test showed that the increase is likely a result of random chance ($t(18) = 0.078$, $p = 0.4695$). Therefore the students in the "stereo" group did not improve because of the training. In contrast, the "mono" group's post-test error ratings ($M = 2.08$, $SD = 1.26$) improved compared to the pre-test error ratings ($M = 2.56$, $SD = 1.04$). This large difference of $-0.472$ have a statistical significance of $p = 0.082$ ($t(17) = 1.45$). This means the students of the "mono" group did improve because of the training in VR. This suggests that students performed better after training in the "mono" condition, which is not the case for the "stereo" condition. To measure the effect of the 3D rendering mode on the learning

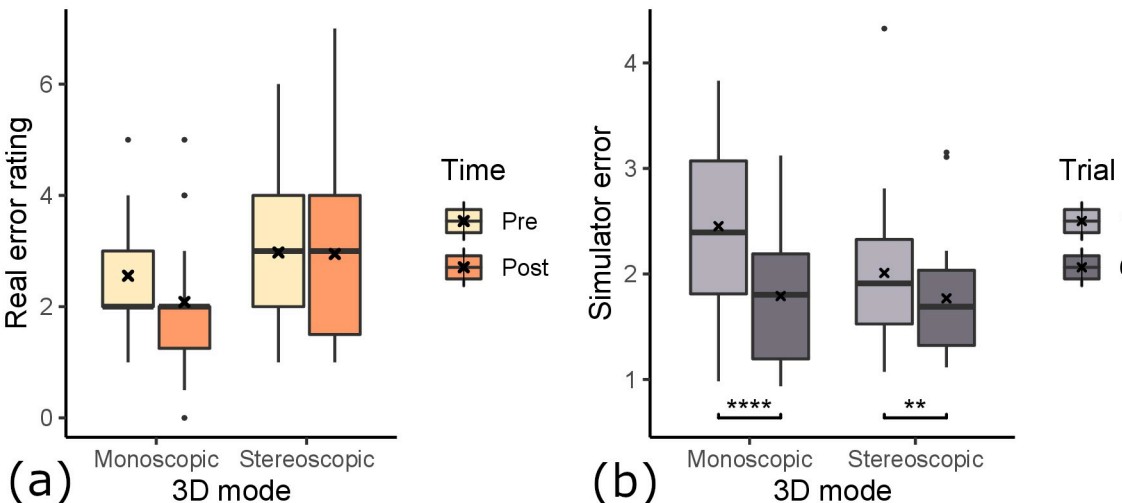

**Fig 9. The effect of 3D rendering mode on learning effect.** (a) Error determined by expert dentists on real outcome. (b) Error determined by algorithm on simulator outcome (**: $p < 0.01$, ****: $p < 0.0001$). For both assessment methods, the monoscopic rendering mode is associated with larger performance improvement.

effectiveness we compared the mean learning gains using a parametric two-tailed t-test. The differences of means of the learning gain between "mono" ($M = -0.472$) and "stereo" ($M = -0.026$) is 0.446, however the difference is not statistically significant ($p = 0.348$).

We also looked at the influence of 3D rendering mode on the in-simulator learning gain (see Fig 9b). A t-test showed no statistically significant difference for the learning gain inside the simulator ($t(34.965) = 1.174$, $p = 0.249$). The simulator error ratings for monoscopic and stereoscopic rendering modes were both improved. However, the simulator learning gain was larger for the monoscopic 3D condition ($M = -0.66$, $SD = 0.90$), similarly to the influence on the real-world learning gain. The training gains in the stereoscopic condition were −0.24 on average ($SD = 0.99$). A t-test revealed that in both conditions, the simulator error ratings were statistically significantly lower after 6 trials compared to the first trial. In the monoscopic condition, the simulator error rating was reduced from 2.45 ($SD = 0.85$) to 1.79 ($SD = 0.62$) ($t(17) = 7.95$, $p < 0.0001$). In the stereoscopic condition, the simulator error rating was reduced from 2.01 ($SD = 0.74$) to 1.77 ($SD = 0.59$) ($t(18) = 2.70$, $p < 0.0001$).

### 5.3 Hand-tool alignment

To determine the impact of hand-tool alignment on participants' performance (see Fig 10), we regard the data of group 1 & 2 as one set of data ("aligned"), and 3 & 4 as the other set of data ("misaligned"), controlling for the stereo factor. The misalignment group did slightly worse on their post-test ($M = 2.86$, $SD = 1.85$), compared to their pre-test ($M = 2.83$, $SD = 1.31$). This small difference of 0.0278 was however shown by the t-test to be likely by random chance ($t(17) = 0.078$, $p = 0.531$). Therefore the participants of the group "misalignment" did not improve by virtual training. However, the "alignment" group improved from their pre-test ($M = 2.71$, $SD = 1.1$) by −0.5 from their post-test ($M = 2.21$, $SD = 1.19$). The t-test shows a statistical significance of $p = 0.0598$ ($t(18) = 1.635$). This suggests that the participants of the "alignment" group improved their error ratings because of the virtual drilling training. This shows, that virtual hand-tool alignment is important for effective training using a virtual simulator.

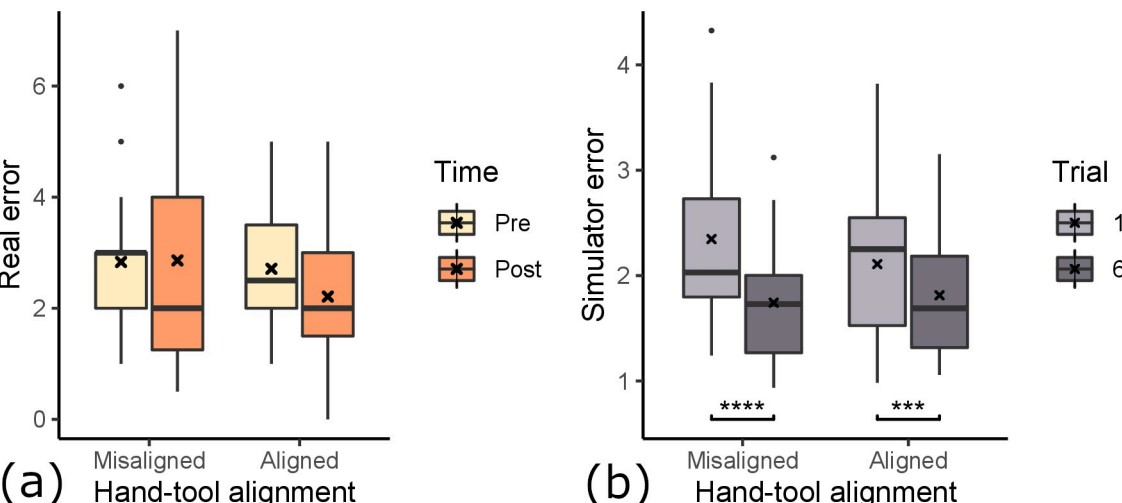

**Fig 10. The effect of hand-tool alignment on learning effect.** (a) Error determined by expert dentists on real outcome. The alignment of hands & tools shows better performance improvement. (b) Error determined by algorithm on simulator outcome (**: $p < 0.01$, ****: $p < 0.0001$), with no noticeable effect.

We also examined the influence of hand-tool alignment on the in-simulator learning gain (see Fig 10b). In both conditions, participants improved their error ratings significantly. The aligned condition improved significantly from 2.11 ($SD$ = 0.76) to 1.81 ($SD$ = 0.61) ($t(18)$ = 4.10, $p < 0.001$). The misaligned condition improved slightly more with significantly lowering of the error from 2.35 ($SD$ = 0.88) to 1.74 ($SD$ = 0.59) ($t(17)$ = 5.09, $p < 0.0001$). We did not find any influence of the hand-tool alignment on the simulator learning gains ($t(33.08)$ = −0.97, $p$ = 0.17). The participants in the aligned condition improved their simulator error rating by an average of −0.60 ($SD$ = 1.05), and the misaligned condition improved on average by −0.30 ($SD$ = 0.87).

## 5.4 Suitability for assessment

Suitability for assessment describes the transfer from previously acquired real psychomotor skills to the simulator. We quantify the suitability by the correlation of pre-training score on plastic teeth and in-simulator performance at the first training session.

When looking at Fig 11a, we can see that the correlation in all samples is very low and insignificant ($R$ = 0.018, $p$ = 0.85). When looking at the factor hand-tool alignment (see Fig 11c), we can see that the aligned condition produces a better skill transfer from pre-training to simulator ($R$ = 0.15, $p$ = 0.25) compared to the misaligned condition, which is even negative ($R$ = −0.12, $p$ = 0.39). When looking at the mean differences in initial simulator performance (see Fig 12b), there is a small difference. The aligned condition resulted in a slightly lower initial simulator error ($M$ = 2.09, $SD$ = 0.74) compared to the misaligned condition ($M$ = 2.39, $SD$ = 0.95) ($t(35.98)$ = 1.107, $p$ = 0.138). Similarly, we see an influence of 3D rendering mode (see Fig 11b) as a factor on the skill transferability from pre-training error to simulator error. Here, stereoscopic 3D had a positive correlation ($R$ = 0.2, $p$ = 0.14), whereas monoscopic 3D had a negative correlation ($R$ = −0.16, $p$ = 0.25). Additionally, there is a statistically significant impact on mean initial simulator performance (see Fig 12a). The stereoscopic 3D condition resulted in significantly lower initial simulator error ($M$ = 1.99, $SD$ = 0.73) compared to the monoscopic 3D condition ($M$ = 2.48, $SD$ = 0.92) ($t(36.09)$ = 1.86, $p < 0.05$). When looking at the four groups (see Fig 11d) with the factors combined, we can see that group 1 (stereoscopic 3D & hand-tool alignment) shows by far the strongest skill transfer correlation, with a moderate, significant correlation ($R$ = 0.41, $p < 0.05$) between pre-training error and simulator error. The other groups either had a low correlation, and group 4 (monoscopic & misaligned) even had a moderate negative correlation ($R$ = −0.39, $p$ = 0.067), meaning students with good real-world skill tended to perform worse in the simulator than those with bad real-world skill.

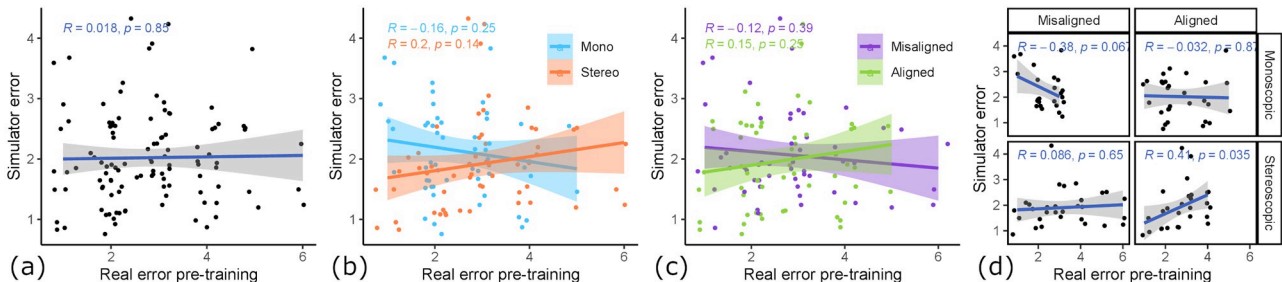

**Fig 11. Relationship between pre-training score and initial simulator score.** The correlation between error rating pre-training (as measured by expert dentists) and initial simulator performance (as measured by simulator error ratings on day 1). (a) All samples, no correlation. (b) Influence of 3D rendering mode. Stereo 3D shows a positive and mono 3D a negative correlation. (c) Influence of hand-tool alignment. Alignment shows a positive and misalignment a negative correlation. (d) Influence of condition groups. The combination of stereo 3D & aligned (group 1) shows a moderate positive correlation.

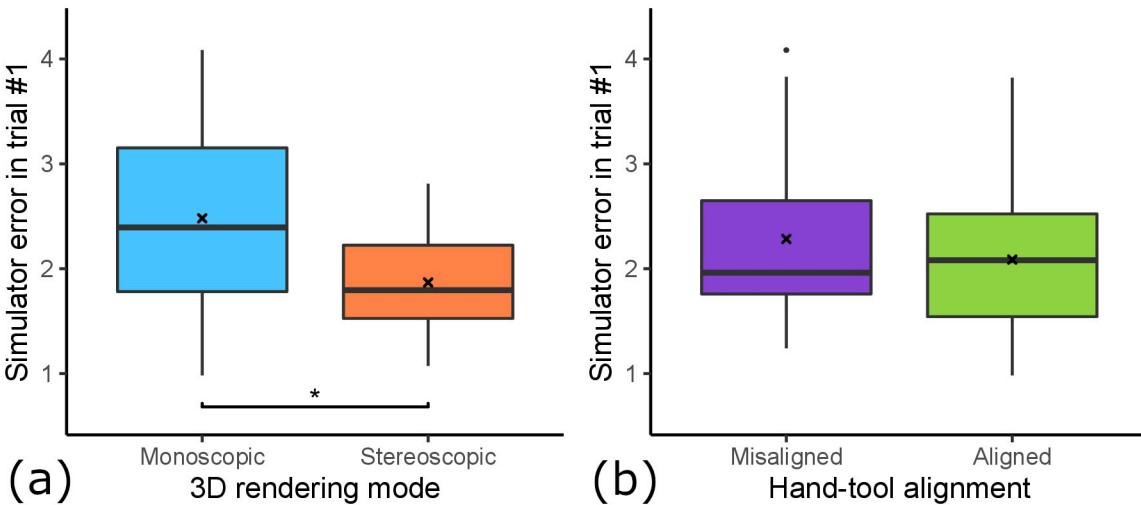

**Fig 12. The influence of both factors on initial simulator performance.** (a) Influence of 3D rendering mode. Stereo 3D shows significantly lower initial simulator performance than mono 3D (*: $p < 0.05$). (b) Influence of hand-tool alignment. Alignment shows lower initial simulator performance than misalignment.

## 5.5 Learning transfer

Learning transfer describes the transfer of psychomotor skills learned inside the simulator to real-world assessed skill. We analyze the factor influences on learning transfer by looking at the correlation of real learning gain to virtual learning gain. Real learning gain is measured by pre- and post-training skill assessment, as rated by expert dentists. Virtual learning gain is measured by looking at the automated error rating of trial #1 and trial #6.

When looking at Fig 13a, there is an overall moderate correlation between simulator gain and training gain ($R = 0.25$, $p = 0.14$). The hand-tool alignment factor had almost no influence on the learning transfer (see Fig 13c), where in the aligned condition, the correlation is similarly high ($R = 0.3$, $p = 0.21$) like in the misaligned condition ($R = 0.27$, $p = 0.27$). However, the 3D rendering mode had a noticeable impact on the learning transfer (see Fig 13b). The monoscopic condition showed a moderate correlation between simulator gain and learning gain ($R = 0.38$, $p = 0.12$), whereas the stereoscopic condition showed almost no correlation

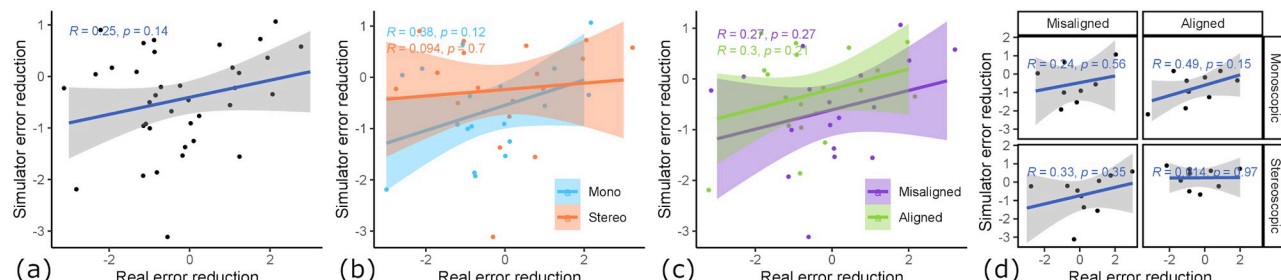

**Fig 13. Relationship between real and virtual learning gain.** The correlation between learning gain as measured by real outcomes (pre- to post-training) vs. learning gain measured by simulator outcomes (trial #1 to trial #6). (a) All samples, moderate correlation. (b) Influence of 3D rendering mode. Mono 3D shows a moderate correlation, while stereo 3D shows no correlation. (c) Hand-tool alignment has no influence. (d) Influence of condition groups. Group 2 (mono 3D & aligned) shows the highest correlation ($R = 0.49$).

($R = 0.094$, $p = 0.7$). Furthermore, looking at the condition combinations (see Fig 13d), we can see that almost all groups showed a positive correlation, except for group 1 showing no correlation ($R = 0.014$, $p = 0.97$). Group 2 had the highest correlation ($R = 0.49$, $p = 0.15$), this indicates that in our setup, the conditions monoscopic 3D & hand-tool alignment create a learning environment that best translates the acquired skill to the real world. Please note that all correlations here are statically insignificant, since we only have 37 total data points which are even less when split up, however, the overall tendency for a positive correlation does not change in any subset of the data.

## 5.6 Eye-tooth distance

We compared the mean eye-tooth distance for participants in both 3D rendering conditions and found a large influence of the 3D rendering mode (see Fig 14a). The monoscopic condition had a significantly lower mean eye-tooth distance ($M = 19.83$, $SD = 8.19$) compared to the stereoscopic condition ($M = 25.68$, $SD = 6.82$) ($t(207.52) = -5.71$, $p < 0.001$).

The hand-tool alignment had a similarly large effect on the mean eye-tooth distance. However, that is easily explained by the fact that we implemented misaligned hands and tools by calibrating with an offset that will result in the virtual tooth being further away from the participant. Therefore they had to move their head closer to the tooth to get the same eye-tooth distance, which some participants did not do. In the hand-tool aligned condition, the mean eye-tooth distance was significantly lower ($M = 19.42$, $SD = 5.99$) compared to the misaligned condition ($M = 26.41$, $SD = 8.47$) ($t(183.9) = 6.97$, $p < 0.001$).

As the learning gain influence of 3D rendering mode was counter-intuitive for us, we suspected that it could be a result of the accommodation near point limitation of stereoscopic rendering in combination with the limited resolution, which resulted in stereoscopic 3D condition having a tooth with effectively lower resolution. To analyze this, we correlated the average eye-tooth distance with the real-world learning gains (see Fig 15). Over all data points, there was a weak but statistically significant positive correlation between mean eye-tooth distance and learning gain ($R = 0.15$, $p < 0.05$). Further analysis showed that if we control for

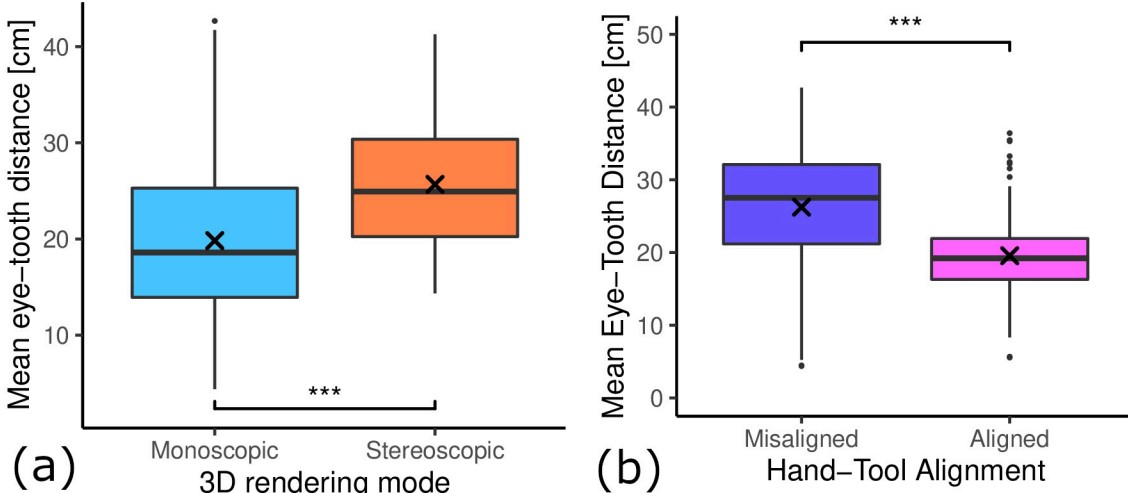

**Fig 14. The influence of both factors on mean eye-tooth distance.** (a) Influence of 3D rendering mode. Mono 3D shows significantly lower eye-tooth distance than stereo 3D. (a) Influence of hand-tool alignment. Alignment shows significantly lower eye-tooth distance than misalignment. (***: $p < 0.001$).

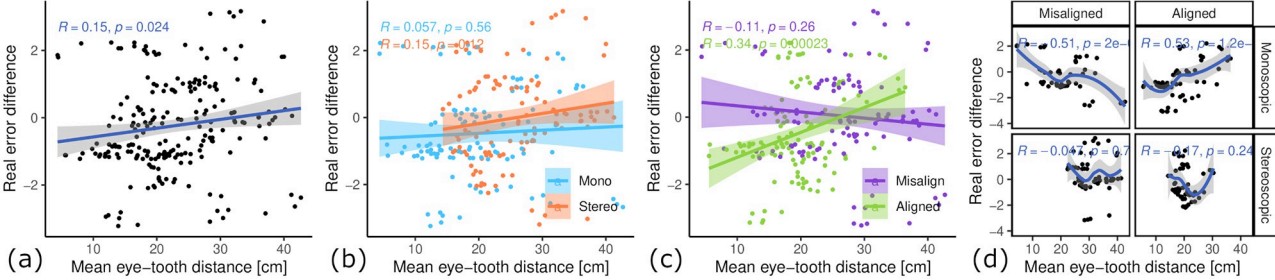

**Fig 15. Relationship of mean eye-tooth distance and learning gains.** (a) All samples show a weak correlation. (b) Influence of 3D rendering mode. Both correlations are weak. (c) Influence of hand-tool alignment. Alignment shows a moderate positive correlation. (d) Influence of condition groups. Group 2 (alignment & mono 3D) shows a strong positive correlation and group 4 (misalignment & stereo 3D) a strong negative correlation. The samples in the mono 3D have a global minimum and maximum at either extremes, whereas stereo 3D has a global minimum in the middle and the performance to the extremes gets worse. This suggests that there is an optimal distance for stereo 3D, a value after which the stereo vision suffers because of the large stereo disparity. For mono 3D (& aligned) the shorter the distance to the tooth, the better the learning performance.

hand-tool alignment, data points in both conditions mono and stereoscopic 3D showed no or only weak correlation with mean eye-tooth distance (see Fig 15b). However, when controlling for 3D rendering mode, the data points in the aligned hands and tools showed a moderate positive correlation with strong statistical significance ($R = 0.34$, $p < 0.001$) (see Fig 15c). The data points in the misaligned condition had a weak negative correlation between eye-tooth distance and learning gain ($R = -0.11$, $p = 0.26$). If we look at the data points inside the aligned condition (see Fig 15d, second column) in the combination with monoscopic 3D (see Fig 15d, second column, first row), there is an even stronger correlation between mean eye-tooth distance and learning gain ($R = 0.53$, $p < 0.0001$). Interestingly, the stereoscopic 3D & aligned group (see Fig 15d, second column, second row) shows a global maximum learning gain at around 23 cm mean eye-tooth distance.

## 6 Discussion

Learning transfer describes the extent to which skill acquisition translates from the acquisition modality to a target modality. In this study the acquisition modality is the simulator and the target modality is performance on realistic plastic teeth, as evaluated by dental experts. To assess the learning transferability of our simulator, we looked at (i) the absolute real learning gain and (ii) the correlation of learning gain and simulator learning gain. We hypothesized both experimental variables, 3D rendering mode and hand-tool alignment, to have a positive impact on the learning transferability of our VR simulator. Our results suggest that stereoscopic 3D had no statistically significant impact on the real-world learning gains (see Fig 9a). However, the mean learning gain was higher for the monoscopic 3D condition, which is the opposite of our hypothesis. We formulated this hypothesis based on our intuition of an additional depth cue increasing performance and the findings of McIntire et al. [16], which reported that 60% of user studies showed that stereoscopic 3D had a positive impact on performance. A more careful consideration of McIntire et al. in hindsight shows that they were focused on in-simulator performance, while we are concerned with learning gain. In fact, when looking at performance, purely measured by simulator error rating on trial #1 (see Fig 12a), stereoscopic 3D had a significant positive impact on user performance, consistent with the findings of McIntire et al. Additionally, McIntire's literature review spans a wide variety of tasks, whereas complex surgery on a small object (like in our user study) is a very uncommon task that puts special requirements on the display, especially resolution. However, we expected

the increased user performance to also translate to increased learning effectiveness. This was also not the case for the virtual learning gains (see Fig 9b). We further investigated the impact of stereoscopic 3D on learning effectiveness by correlating the virtual and real learning gains across all conditions, thereby controlling for the spread in virtual learning gains (see Fig 13). We found that the overall correlation was moderate, which suggests that participants that increased their simulator score also tended to increase their real-world score. Stereoscopic 3D had a negative impact on the correlation, compared to the monoscopic 3D condition. In fact, when comparing the learning correlations of group 1 (stereo & aligned) and group 2 (mono & aligned), group 2 has a strong correlation, whereas group 1 has none. This also suggests that skills learned inside the simulator in monoscopic 3D translate better to the real-world. Our use-case involves looking at a small object to make out fine details. Therefore we also recorded and analyzed eye tracking data (see Fig 14), which shows a significant impact of 3D rendering mode on mean eye-tooth distance, with users of monoscopic 3D having a significantly lower mean distance compared to stereoscopic 3D. Interestingly, we also noted a much lower standard deviation in the stereo 3D condition and a global minimum at around 15 cm that is larger than the expected near point at 9.92 cm [28], which suggests an optical lower bound in the stereoscopic 3D rendering. By correlating the mean eye-tooth distance per trial to the real-world learning gain, we found that in group 1 (stereo & aligned) the optimal learning gain is achieved in the middle of the distribution, at 23 cm distance, whereas this optimum is located at the extremes for other groups. By our estimation, the near-point in the simulator with stereoscopic 3D is located at the same distance of 23 cm. We suspect the user is trying to be as close to the tooth as possible to maximize the resolution of the tooth, while also being far enough away to be able to focus the tooth. In fact, group 2 (mono & aligned) showed a strong linear relationship between eye-tooth distance and real-world learning gain, which shows that the distance explains over 50% of the learning gains, as it results in a higher tooth resolution on screen. This intuitively makes sense, as there is no perceivable near-point in the monoscopic 3D condition and participants can essentially look as close to the tooth as they like. They thereby increase the tooth resolution on screen and receive more information, which could be regarded as immediate feedback of the drilling procedure, as they could see more details. Paricipants in the stereo condition did not not have a chance to receive this form of immediate feedback. As it has been shown many times, timely feedback has a significant positive impact on learning effectiveness compared to delayed feedback [33–35]. Thus, our first hypothesis $H_{V_{learn}}$ could not be confirmed. However, it is likely that eye-tooth distance is a confounding variable that explains the counter-intuitive influence of 3D rendering mode on learning gain. Future studies that incorporate small objects in VR should control their stereopsis to allow for a near-point that is realistic for the target task [28]. We suspect that when controlling for the tooth resolution in the described manner, stereoscopic 3D could have a positive impact on learning effectiveness of a VR simulator.

Hand-tool alignment had a positive impact on the learning effectiveness of the simulator, with higher real-world learning gains in the alignment condition (see Fig 10). This confirms our hypothesis $H_{A_{learn}}$. However, when correlating virtual learning gains and real-world learning gains (see Fig 13), we found no significant impact from hand-tool alignment, similar to the 3D rendering mode. Both conditions, aligned and misaligned, showed a moderate correlation between virtual and real learning gains, meaning both conditions translate the learned skills similarly to the real world. This indicates, that in the aligned condition, participants that did not improve substantially in the simulator scoring still tended to improve at the real task, which was not the case for the misaligned condition. In fact, we could see a slightly larger simulator learning gain for the misaligned condition (see Fig 10b). Based on these findings, we

showed that a simulator with hand-tool misalignment, such as when using a desktop monitor, is more likely to have weak learning transfer. Users of these kinds of simulators could be more likely to learn the intricacies of the simulator, not the real task.

Skill assessment is the process of determining a person's skill in a certain task or field, compromised of a set of tasks. Often, this skill is the foundation to determine if a person has also acquired expertise in this task or field. It is essential for the assessment tool to have accurate and reliable skill evaluation. To determine the feasibility of our simulator as a skill assessment tool, we looked at the correlation between the pre-training error, which is the ground truth of the student's current skill level, and the simulator error ratings on day 1. We hypothesized that both variables would positively impact the skill transfer from real-world to the VR simulator. Many studies find stereoscopic 3D to have a positive impact on performance in virtual surgical tasks [20, 21], which led us to hypothesize stereoscopic 3D would also have a positive impact on skill transfer. This follows the logic that higher virtual performance indicates intuitive usability of the simulator, which should better translate real-world experience to simulator experience. A simulator with intuitive usability would help identify the simulator's suitability as an automated and objective skill assessment tool, which is something the medical community is looking for [9, 10]. To analyze the intuitive usability, we mainly considered the trials on day 1 of the virtual training, as the data shows a learning curve that starts plateauing on day 2. When looking at the simulator scores on trial #1 (see Fig 12), we found a significant positive impact of stereoscopic 3D on the score of trial #1. We further correlated the pre-training real-world score with the simulator score on day 1 (see Fig 11). Here, we found that the 3D rendering mode had an impact on the correlation, with stereo 3D showing a moderate positive correlation, while mono 3D showed a moderate negative correlation. These findings confirm our third hypothesis $H_{V_{assess}}$, that stereoscopic 3D has a positive impact on skill transfer.

We expected hand-tool alignment to have a similar effect. However this is mostly based on our intuition, as we did not find studies that deal with this issue. The simulator error in trial #1 was only slightly lower in the aligned condition compared to the misaligned condition (see Fig 12). Interestingly, when correcting for the real-world skill by correlating pre-training error and simulator error on day 1, we found that hand-tool alignment had a positive impact. Although the impact is lower than the effect of the 3D rendering mode, it still shows that aligned hands and tools improve skill transfer as particpants with low pre-training error tended to also have low simulator error on day 1 in this condition. Therefore, our data suggest that our last hypothesis $H_{A_{assess}}$ is confirmed.

In fact, when looking at both variables together, the effect accumulates. Resulting in the skill correlation being the highest for the group 1 samples (stereo & aligned), with the initial simulator performance correlating over 40% with the expert pre-training assessment. This confirms that this setting is the most intuitive one, as it best translates users' already predominant preparation skill. We can even see the simulator being the basis of development for an automated, reliable and objective skill assessment tool in this setting.

The connection of our simulator and reality is very interesting to look at. Previous studies [20, 21] examined performance and learning differences in dental simulators with stereoscopic and monoscopic rendering. In those studies the task was carried out on simulated geometric objects. Evaluation of skill was done within the simulator, with automated scoring based on material removed. By contrast, our study used the endodontic task of root canal access opening. Evaluation of learning gains was done using pre- and post-testing on realistic plastic teeth, with scoring done by dental instructors using the standard method used in clinical teaching. Thus, it can be argued that our study is done in a more realistic setting and includes evaluation of transferability of learned skills. Transferability is important to evaluate since it is entirely

possible to attain a high level of skill in a simulator, yet not have this in-simulator skill translate to real-world tasks.

## 7 Conclusion

This is the first study to analyse the effect of different aspects of VR realism on transferability of dental skills from VR simulation training to real-world tasks and vice versa. We have found that the alignment of the physical and virtual tools had a positive impact on students' learning gains, compared to students with misaligned physical and virtual tools. Hand-tool alignment was also helpful in increasing simulator usability, suggesting it is easier to adapt to the simulator and is better suited for skill assessment.

Surprisingly, we observed that in our setting monoscopic 3D rendering provided students with more helpful training compared to stereoscopic 3D, as their learning gain was higher. Although it must be noted that our limited sample size did not yield statistical significance. However, this counter-intuitive finding might be confounded by the eye-tooth distance, which was found to be significantly lower for the monoscopic 3D condition. Therefore, future studies need to control for eye-tooth distance, for example by enforcing a similar lower bound in the monoscopic condition, since such a lower bound naturally exists for stereo vision. The stereo vision near point should also be controlled, as we found the near point inside a VR HMD to be larger than in the real world. One possible explanation for this is the limited field of view of HMDs restricting reference points common for both eyes at high inter-ocular disparity. However, despite the large near point, stereoscopic 3D had a significantly better skill transfer, as it showed a high correlation with participants pre-training score. This shows that it is easier for participants to manifest their real-world skill inside the simulator when using stereo 3D. Consequently, it is the desired rendering mode when using a simulator for skill assessment purposes.

## Author Contributions

**Conceptualization:** Maximilian Kaluschke, Myat Su Yin, Peter Haddawy, Siriwan Suebnukarn, Gabriel Zachmann.

**Data curation:** Maximilian Kaluschke, Myat Su Yin.

**Formal analysis:** Maximilian Kaluschke.

**Funding acquisition:** Myat Su Yin, Peter Haddawy, Siriwan Suebnukarn, Gabriel Zachmann.

**Investigation:** Maximilian Kaluschke.

**Methodology:** Maximilian Kaluschke, Myat Su Yin, Siriwan Suebnukarn, Gabriel Zachmann.

**Project administration:** Peter Haddawy, Siriwan Suebnukarn, Gabriel Zachmann.

**Resources:** Myat Su Yin, Peter Haddawy, Gabriel Zachmann.

**Software:** Maximilian Kaluschke.

**Supervision:** Myat Su Yin, Peter Haddawy, Siriwan Suebnukarn, Gabriel Zachmann.

**Validation:** Maximilian Kaluschke, Myat Su Yin, Peter Haddawy, Siriwan Suebnukarn, Gabriel Zachmann.

**Visualization:** Maximilian Kaluschke.

**Writing – original draft:** Maximilian Kaluschke, Myat Su Yin, Peter Haddawy, Siriwan Suebnukarn, Gabriel Zachmann.

Writing – review & editing: Maximilian Kaluschke.

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
