## [Decision Letter · Decision Letter 0]

30 Jan 2023

PONE-D-22-35513The effect of 3D stereopsis and hand-tool alignment on learning effectiveness and skill transfer of a VR-based simulator for dental trainingPLOS ONE

Dear Dr. Kaluschke,

Thank you for submitting your manuscript to PLOS ONE. After careful consideration, we feel that it has merit but does not fully meet PLOS ONE’s publication criteria as it currently stands. Therefore, we invite you to submit a revised version of the manuscript that addresses the points raised during the review process. The reviewers agreed that the manuscript and associated study had strong merit. However, Reviewer #2 raised concerns, which I agree with, regarding the clarity of the objectives and hypotheses of the work, the description of the study design and outcomes variables used, and the way in which the results and discussion are presented. With this in mind, it is necessary for you to revise the manuscript in response to the comments of the reviewers (especially those raised by Reviewer #2) in order for this manuscript to be acceptable for publication.  

We look forward to receiving your revised manuscript.

Kind regards,

Joshua William Giles, Ph.D.

Academic Editor

PLOS ONE

Journal Requirements:

3. hank you for stating the following in the Acknowledgments Section of your manuscript: 

"This work was partially supported by a grant from the Mahidol University Office of

International Relations to Haddawy in support of the Mahidol-Bremen Medical Informatics Research Unit (MIRU), by a fellowship from the Hanse-Wissenschaftskolleg Institute for Advanced Study, and by a Young Researcher grant from Mahidol University to Su Yin."

4. We noted in your submission details that a portion of your manuscript may have been presented or published elsewhere.

"The data analysis on real-world learning gains was published in the IEEE ICHI conference. This excludes the automated scoring algorithm, the simulator score analyses and eye tracking data analysis."

Reviewers' comments:

Reviewer's Responses to Questions

**Comments to the Author**

1. Is the manuscript technically sound, and do the data support the conclusions?

Reviewer #1: Yes

Reviewer #2: Yes

2. Has the statistical analysis been performed appropriately and rigorously? 

Reviewer #1: Yes

Reviewer #2: Yes

3. Have the authors made all data underlying the findings in their manuscript fully available?

Reviewer #1: Yes

Reviewer #2: Yes

4. Is the manuscript presented in an intelligible fashion and written in standard English?

Reviewer #1: Yes

Reviewer #2: Yes

5. Review Comments to the Author

Reviewer #1: Dear Authors,

I read with great interest your manuscript on "The effect of 3D stereopsis and hand-tool alignment on learning effectiveness and skill transfer of a VR-based simulator for dental teaching", which analyzes the effects of 3D rendering modes, hand-tool alignment, transferability to the clinical situation, as well as the effects of eye-to-tooth distance (accommodation near point). The main findings of the manuscripts are that hand-tool alignment seems to have a positive effect, while stereoscopic 3D rendering has a negative effect on learning effectiveness, although it improves accuracy, which could be influenced by eye-to-tooth distance.

The manuscript describes in great detail the development of the simulator and provides insight into the controversial topic of whether the effect of stereoscopic 3D is beneficial or counterproductive.

The manuscript is well written, the results are adequately presented and briefly discussed. This manuscript may provide a good basis for further investigation of these findings.

I only have some minor comments

Minor comment:

- Fig. 8, 11, 13 and 15 have a low resolution and are difficult to read

- rather low sample size (10 participants per group)

- age of the participants could be of interest (since you refer to the accommodation near point of the participants during the eye tracking analysis)

- Please recheck and rephrase sentence lines 467 - 469

Reviewer #2: *Overall comments:

An interesting premise for a study, and a well-developed system in which the authors combined VR and haptics to create a dental surgical simulator. However, the paper needs significant reworking of its structure. It is unclear throughout the introduction (and by extension, the rest of the paper) what the research objectives and hypotheses of this study were. What gap in the literature and in the state of the art is this study filling? What are the contributions? The discussion is very short, and the results section is quite bloated, as there is much overlap between the two. A restructuring of this paper should clearly highlight and justify what outcome measures were taken before the results section. Use the results section to present and initially discuss the results themselves and discuss the statistical methods employed. However, the implications and deeper analysis of the results should be happening in the Discussion section.

Some technical aspects need clarification: How have the authors calibrated the haptic device with the VR tracking system? The measure of ‘eye-to-tooth’ distance is introduced in the results section, this needs to come much earlier, and the motivation for taking this measure should have been introduced. A key point to flag here is the idea of variable interpupillary distance (IPD), this needs to be rephrased, as it gives the impression that interpupillary distance can change.

The discussion and conclusion read like a reduced version of the results. Consider showing the reader what the key set of findings are. What are the main contributions, and clearly highlight the study’s limitations here. Plenty of care has been taken to present the statistical analysis of the results, however, some more clarity is needed in the figures and graphs to more clearly highlight the subject groupings. After addressing these points, the abstract needs to be adapted to highlight these changes. There are minor spelling and grammar mistakes throughout the paper, which have been highlighted, but the authors should carefully revisit the manuscript to remove any other errors.

*Introduction

Line 8: ‘so-called’ spelling

Line 15 onward: ‘concrete benefits of this approach’, ‘real-world learning gains’, ‘real-world learning effects’ such as?

Line 32: When discussing this research goal, the authors should provide a definition or clarification of what the ‘simulation aspects’ of VR technologies are

Line 43: ‘provides valuable information’ spelling mistake. Also this statement is very broad, what is the valuable information your study provides? And is this in the right place in the paper? Why make reference to results in the introduction when the reader has no context of what the experiments were or the results themselves?

Introduction is lacking: a clear definition of research/study objectives, the hypothesis, the motivation, and the broader contribution of this study.

*Related Work

77: A description of another group’s work, but possibly a more representative description of what the authors have done too. IPD wasn’t changed, but the authors manipulated the HMD system to produce similar images in the left and right eyes?

Throughout this section the reader is being directed that rendering in a more 2D style and removing stereoscopic effects has negative effects, so this begs the question, why did the authors choose to add this to the system and study? The authors need to justify the inclusion or the motivation more clearly behind doing so.

Whilst the section does well to highlight other works and to explain those contributions, this section would benefit from the authors providing context on where their own system fits in to the state of the art. For example, after looking at the other works described in this Section, are the authors exploring other avenues of research which previous works have not covered? Is the system/setup being proposed in this study significantly different in principle from those highlighted in this section? If so, how?

*Simulator

154: The setup has been tuned to mimic plastic phantom models as opposed to real teeth. The justification for this decision appears much later in the manuscript, simply shift that up here instead

*Stereo Rendering:

166: There is some confusing phrasing and use of IPD (inter-pupillary distance) in this section and in the remainder of the paper. IPD is traditionally used to refer to a physical quantity, the distance of between the user’s pupils. My impression from reading this excerpt of the paper is that the system here scales the user and entire environment up with respect to real world units, and thus leverages how the images are rendered to the user by the HMD. Due to the stereo separation effects, the images in the left and right eye appear more similarly to each other than they would if the scale of objects had remained untouched. If this is what the authors’ system set out to do, then this needs to be made a little clearer. Additionally referring to it as stereo/mono rendering may cause some confusion, but it is highly recommended to rethink the use of ‘effective IPD’ which to my knowledge isn’t a standardized term.

*Hand-Tool Alignment

From reading this section, it appears that the haptic device (Phantom) doesn’t need to be dynamically tracked by the HMD or its tracking system as the Phantom and the HMD’s lighthouse tracking system are rigidly fixed with respect to each other. However, was some form of calibration done to align the two frames of reference with respect to each other so you can correctly transform motion from the Phantom’s coordinate frame into the VR frame?

185: The haptic devices were moved in 20 cm and 50 cm, in what coordinate frame? With respect to the table? Can the authors clearly label these dimensions in the relevant figure to avoid ambiguity as to which setup is the ‘misaligned’ one?

*3.3 Automated Outcome Scoring

192: Define first use of this symbol, which appears to be Cohen’s Kappa

242: How were these voxel classifications achieved?

*4 User Study:

262: Condition (ii) for exclusion: removing those who achieved less than 70% in assessment tests. Can the authors justify why this was done?

270: This is a valid justification for the use of phantom teeth in the study, move it earlier in the manuscript where it is first discussed.

282: 5-30 minutes is a very large range for elapsed time. Is this a minimum and maximum? Useful to include some summary statistics here regarding the trial times.

*5 Results

303: Fig 7 Caption spelling mistake: ‘consecutive trials are compared’

Also Fig 7B itself is a little unclear with respect to the groupings. The flowchart helps understand the breakdown of the trials, but the graph itself or the caption underneath can benefit from more clearly highlighting the breakdown of the groups

460: grammar: ‘This confirms’ and line 461: “to assess one’s skill”.

*5.5 Eye tracking

This is the very first mention of eye-tracking in the manuscript. There has been no leadup in the introduction or the methods. This setup should not appear for the first time in the results section, the motivation for recording eye-tracking needs to be brought to the reader’s attention much earlier.

Additionally, is it appropriate to title this eye tracking or referring to it as eye-to-tooth distance? Gaze and eye-tracking are different concepts. From reading this section, it appears you use the pose returned by the HMD to estimate the gaze of the user. How is this used to calculate the eye-to-tooth distance? Does the HMD’s position overlap with the user’s eye? A clear definition of this distance is needed, or an illustration, if space permits.

Line 490: ‘Modern HMDs’ grammar

Line 500: The authors recorded motion over an entire trial and formulated a single data point from this. How was the single data point calculated, was the mean distance taken? Also, if recording motion over an entire trial, how do you prevent recording any anomalous motion/consistently across users? Did the authors only start and end recording at specific points in the procedure?

510: ‘for a healthy..’ spelling

*Discussion/Conclusion:

Some element of summarizing the study’s origin and motivations is needed here. The discussion reads as a conclusion of all the discussion that happened in the results section here. It is far too short to cover the system aspects and study design of this paper.

Line 574: Once again, the motivation of measuring ‘eye-to-head’ data is missing, leaving a reader to question why this was measured in the first place. Also it’s a misnomer to call it ‘eye tracking’ if that isn’t what was done in this study.

Currently, the discussion offers little contribution to the manuscript in several respects. Currently, it describes the results at a more basic level than the Results section itself. To make it a more meaningful discussion, the authors should discuss at the very least: (a) the implication of the results on the future of simulator based or VR-based training for dentistry (b) how do the results compare to those in other comparable studies (c) what are the other potential challenges to more widespread adoption of such systems or measurement techniques as discussed by the authors for procedures in a similar sphere (dental and/or maxillofacial procedures perhaps?)

The discussion also needs to clearly highlight study limitations and future areas of work. The conclusion is missing a concise set of highlights or takeaways from this paper. Once again, as with the introduction, it should be emphasized what gap in literature this fills, and what the key contributions of this study are in comparison to the rest of the space.

6. PLOS authors have the option to publish the peer review history of their article (what does this mean?). If published, this will include your full peer review and any attached files.

Reviewer #1: **Yes: **Dr. Christian Diegritz

Reviewer #2: No

---

## [Author Response · Author response to Decision Letter 0]

27 Jun 2023

Review 1

• Fig. 8, 11, 13 and 15 have a low resolution and are difficult to read

o Please excuse the low resolution, that is a result of the submission pdf generation, the images are available as vector graphics, so I could generate any resolution, and I generated the rasterized the resolution according to the submission guidelines. For your convenience I can provide a PDF with the graphs included in higher resolution, if the system will allow me. For your convenience I uploaded a version with Figures in PDF: https://www.dropbox.com/s/n5jhq06qwgjur3n/Manuscript_with_Figures.pdf?dl=0

• rather low sample size (10 participants per group)

o Yes, that is a downside of the study design and the found statistical significances could have been even better if the sample size was larger.

• age of the participants could be of interest (since you refer to the accommodation near point of the participants during the eye tracking analysis)

o Unfortunately, we did not record the ages. However, as all participants are from the same course and year, the age range is 20-24. We updated the near point average with the average for ages 20-25, as that is more appropriate.

• Please recheck and rephrase sentence lines 467 - 469

o We corrected the mistake.

Review 2

Overall Feedback

• An interesting premise for a study, and a well-developed system in which the authors combined VR and haptics to create a dental surgical simulator. However, the paper needs significant reworking of its structure. It is unclear throughout the introduction (and by extension, the rest of the paper) what the research objectives and hypotheses of this study were. What gap in the literature and in the state of the art is this study filling? What are the contributions? The discussion is very short, and the results section is quite bloated, as there is much overlap between the two. A restructuring of this paper should clearly highlight and justify what outcome measures were taken before the results section. Use the results section to present and initially discuss the results themselves and discuss the statistical methods employed. However, the implications and deeper analysis of the results should be happening in the Discussion section.

o We formulated four major hypotheses, which we had going into the study, at the end of the introduction. We extended the Discussion chapter to go into more depth on the findings and shortened the Results chapter by removing points that are now addressed in the Discussion.

• Some technical aspects need clarification: How have the authors calibrated the haptic device with the VR tracking system? The measure of ‘eye-to-tooth’ distance is introduced in the results section, this needs to come much earlier, and the motivation for taking this measure should have been introduced. A key point to flag here is the idea of variable interpupillary distance (IPD), this needs to be rephrased, as it gives the impression that interpupillary distance can change.

o The chapter "Hand-Tool Alignment" describes the calibration procedure, which we now extended slightly to be clearer, and we also added the calibration offset values of our exact setup.

• The discussion and conclusion read like a reduced version of the results. Consider showing the reader what the key set of findings are. What are the main contributions, and clearly highlight the study’s limitations here. Plenty of care has been taken to present the statistical analysis of the results, however, some more clarity is needed in the figures and graphs to more clearly highlight the subject groupings. After addressing these points, the abstract needs to be adapted to highlight these changes. There are minor spelling and grammar mistakes throughout the paper, which have been highlighted, but the authors should carefully revisit the manuscript to remove any other errors.

o We have included contributions of our study that follow the train of thought in the Discussion and updated the abstract.

Introduction

• Line 8: ‘so-called’ spelling

o We corrected the spelling.

• Line 15 onward: ‘concrete benefits of this approach’, ‘real-world learning gains’, ‘real-world learning effects’ such as?

o The concrete benefits refer to the rest of the paragraph. Real-world learning gains and effects refers to the transfer from virtual to real-world learning for the specific surgical procedure, as measured by real-world assessment. We clarified this in the manuscript.

• Line 32: When discussing this research goal, the authors should provide a definition or clarification of what the ‘simulation aspects’ of VR technologies are

o The two aspects are 3D stereoscopic rendering and hand-tool alignment, which we mention in the begging of the paragraph. We modified the phrasing to be clearer there.

• Line 43: ‘provides valuable information’ spelling mistake. Also this statement is very broad, what is the valuable information your study provides? And is this in the right place in the paper? Why make reference to results in the introduction when the reader has no context of what the experiments were or the results themselves?

o Provide refers to the plural subject "results", so I believe the spelling is correct. Our idea was to shortly summarize the content and contribution of the paper and hint at the results to peak the interest of the reader. However, as it seems out of place, I removed the last sentence that references the results.

• Introduction is lacking: a clear definition of research/study objectives, the hypothesis, the motivation, and the broader contribution of this study.

o We added research objectives and four hypotheses to the end of the introduction. I am a bit confused regarding the want for mentioning of contribution, though the previous comment mentioned it was the wrong place to put the broader contribution that we previously had there.

Related Work

• 77: A description of another group’s work, but possibly a more representative description of what the authors have done too. IPD wasn’t changed, but the authors manipulated the HMD system to produce similar images in the left and right eyes?

o This is a simplified explanation of what we have done. We actually changed the virtual IPD to be nearly 0, which results in both eyes receiving nearly the same image. However, this is technical limitation of the game engine that we used and we don't want to confuse readers with this term so we changed our explanation to this simplified one. We also noted in line 77 that we essentially use the same method to achieve monoscopic 3D.

• Throughout this section the reader is being directed that rendering in a more 2D style and removing stereoscopic effects has negative effects, so this begs the question, why did the authors choose to add this to the system and study? The authors need to justify the inclusion or the motivation more clearly behind doing so.

o There are several studies that come to the conclusion that stereoscopic 3D is superior to monoscopic 3D in regards to user performance and learning effectiveness. However, a considerable number of studies come to the conclusion that it had no or marginal impact. Additionally, most studies used 2D and 3D monitors, not HMDs. The goal of our study was to investigate the impact of the VR technology used, and a major feature of HMDs is stereoscopic vision when compared to 2D monitors. Compared to 3D monitors, hand-tool alignment is the notable feature that HMDs have over them. These features are advertised as important, which we wanted to investigate in detail. As such, we feel stereoscopic 3D is an essential variable to include in such a study. We extended the motivation we give in the Introduction's last two paragraphs and relevant parts of the Related Work to better convey this.

• Whilst the section does well to highlight other works and to explain those contributions, this section would benefit from the authors providing context on where their own system fits in to the state of the art. For example, after looking at the other works described in this Section, are the authors exploring other avenues of research which previous works have not covered? Is the system/setup being proposed in this study significantly different in principle from those highlighted in this section? If so, how?

o We added several summaries of how each related work relates to our study.

Simulator

• 154: The setup has been tuned to mimic plastic phantom models as opposed to real teeth. The justification for this decision appears much later in the manuscript, simply shift that up here instead

o We moved the justification here.

Stereo Rendering

• 166: There is some confusing phrasing and use of IPD (inter-pupillary distance) in this section and in the remainder of the paper. IPD is traditionally used to refer to a physical quantity, the distance of between the user’s pupils. My impression from reading this excerpt of the paper is that the system here scales the user and entire environment up with respect to real world units, and thus leverages how the images are rendered to the user by the HMD. Due to the stereo separation effects, the images in the left and right eye appear more similarly to each other than they would if the scale of objects had remained untouched. If this is what the authors’ system set out to do, then this needs to be made a little clearer. Additionally referring to it as stereo/mono rendering may cause some confusion, but it is highly recommended to rethink the use of ‘effective IPD’ which to my knowledge isn’t a standardized term.

o We simplified this subsection significantly. The "virtual IPD" (the term IPD is often used in computer science to refer to the projection parameter, not the physical quantity) was just a way for us to implement monoscopic rendering, a technical detail, that is not important enough to warrant confusing the reader, so we removed all mentions of IPD. Instead, we simply state that monoscopic rendering was achieved by displaying identical images for both eyes.

Hand-Tool Alignment

• From reading this section, it appears that the haptic device (Phantom) doesn’t need to be dynamically tracked by the HMD or its tracking system as the Phantom and the HMD’s lighthouse tracking system are rigidly fixed with respect to each other. However, was some form of calibration done to align the two frames of reference with respect to each other so you can correctly transform motion from the Phantom’s coordinate frame into the VR frame?

o Yes, we needed to calibrate the systems, this was achieved by having the VR controller located at a fixed offset relative to the haptic devices. During the application running, we can initiate a calibration with the press of a button, which will then read the haptic device positions based on the VR controller position and move the VR camera origin by the difference of virtual and real tool origin, thereby aligning them. We made the explanation in the manuscript clearer.

• 185: The haptic devices were moved in 20 cm and 50 cm, in what coordinate frame? With respect to the table? Can the authors clearly label these dimensions in the relevant figure to avoid ambiguity as to which setup is the ‘misaligned’ one?

o Yes, the directions of down and forward are in relation to the table. We added labels in the figure to make the distinction clearer.

Automated Outcome Scoring

• 192: Define first use of this symbol, which appears to be Cohen’s Kappa

o We just want to report the agreement of the two experts, we don't need the value of Kappa later on.

• 242: How were these voxel classifications achieved?

o We use inner spheres to represent the inner volume of the tooth (Details in "Realistic Haptic Feedback for Material Removal in Medical Simulations") and simultaneously keep a voxel grid which we extract from the inner spheres by defining an implicit surface function and discretizing it with marching cubes at runtime. As this is computationally heavy, we needed to implement spatio-temporal optimizations. Overall, describing the voxel and geometry extract would extend the paper by one or more pages, so we omitted it. I added three sentences in the manuscript to give a similar overview of the extraction process.

User Study

• 262: Condition (ii) for exclusion: removing those who achieved less than 70% in assessment tests. Can the authors justify why this was done?

o Endodontic access cavity preparation requires relevant background knowledge such as anatomy, pathology and physiology before practicing to develop psychomotor skill with simulated teeth. We included a shortened remark in the manuscript.

• 270: This is a valid justification for the use of phantom teeth in the study, move it earlier in the manuscript where it is first discussed.

o We moved the justification for the target feeling to Section 3.

• 282: 5-30 minutes is a very large range for elapsed time. Is this a minimum and maximum? Useful to include some summary statistics here regarding the trial times.

o We included summary statistics on the time inside the simulator for day 1 and 2. We found the times were significantly shorter on day 2. We added the finding to the manuscript.

Results

• 303: Fig 7 Caption spelling mistake: ‘consecutive trials are compared’

o We fixed the mistake.

• Also Fig 7B itself is a little unclear with respect to the groupings. The flowchart helps understand the breakdown of the trials, but the graph itself or the caption underneath can benefit from more clearly highlighting the breakdown of the groups

o We highlight in the caption that there was a time gap between day 1 and day 2.

• 460: grammar: ‘This confirms’ and line 461: “to assess one’s skill”.

o We corrected the mistakes.

Eye tracking

• This is the very first mention of eye-tracking in the manuscript. There has been no leadup in the introduction or the methods. This setup should not appear for the first time in the results section, the motivation for recording eye-tracking needs to be brought to the reader’s attention much earlier.

o We included a mention in the beginning of section 3 on eye-tracking hardware and software used and moved motivation and details in a new subsection under section 3.

• Additionally, is it appropriate to title this eye tracking or referring to it as eye-to-tooth distance? Gaze and eye-tracking are different concepts. From reading this section, it appears you use the pose returned by the HMD to estimate the gaze of the user. How is this used to calculate the eye-to-tooth distance? Does the HMD’s position overlap with the user’s eye? A clear definition of this distance is needed, or an illustration, if space permits.

o Before, we used the term HMD position mistakenly instead of cyclops eye (the center between both eyes). The HMD pose is only used to transform the eye gaze origin from HMD space to world space. The eye-to-tooth distance is the distance from the cyclops eye to the tooth. We improved the description in the manuscript and renamed the Results subsection “Eye-Tooth Distance”. We hope it is clearer now.

• Line 490: ‘Modern HMDs’ grammar

o We fixed the mistake.

• Line 500: The authors recorded motion over an entire trial and formulated a single data point from this. How was the single data point calculated, was the mean distance taken? Also, if recording motion over an entire trial, how do you prevent recording any anomalous motion/consistently across users? Did the authors only start and end recording at specific points in the procedure?

o We calculated the mean over those data points in which the user was looking at the tooth. This is a stable and conservative requirement, because for that the eye tracking has to work and recognize the user's eyes and their gaze ray has to hit the tooth. Therefore, the time when we prepare the user or the HMD was not being worn yet or the user was orientating himself in the VR space are disregarded. The statistical analysis also points to the differences not being a result of randomness, such as measurement noise from anomalous motion.

• 510: ‘for a healthy..’ spelling

o We fixed the mistake.

Discussion/Conclusion

• Some element of summarizing the study’s origin and motivations is needed here. The discussion reads as a conclusion of all the discussion that happened in the results section here. It is far too short to cover the system aspects and study design of this paper.

o We better introduced the study motivations in the Discussion and extended the chapter by moving conclusions of results out of the Results chapter to the Discussion and going more into depth on the implications of findings.

• Line 574: Once again, the motivation of measuring ‘eye-to-head’ data is missing, leaving a reader to question why this was measured in the first place. Also it’s a misnomer to call it ‘eye tracking’ if that isn’t what was done in this study.

o We now motivated the inclusion of eye tracking in the “Visual Perception” section, as well as better explaining the implications of the eye tracking results in the Discussion. We do use in fact eye tracking, but evaluate only a limited amount of the data, as the accuracy is not sufficient to evaluate participants gaze directions in detail.

• Currently, the discussion offers little contribution to the manuscript in several respects. Currently, it describes the results at a more basic level than the Results section itself. To make it a more meaningful discussion, the authors should discuss at the very least: (a) the implication of the results on the future of simulator based or VR-based training for dentistry (b) how do the results compare to those in other comparable studies (c) what are the other potential challenges to more widespread adoption of such systems or measurement techniques as discussed by the authors for procedures in a similar sphere (dental and/or maxillofacial procedures perhaps?)

o We added discussions on (a) implications of the results for future studies and (b) the relation to other comparable studies (when applicable, as there are no comparable studies on hand-tool alignment). We included implications of our results for (c) widespread adoption of similar systems.

• The discussion also needs to clearly highlight study limitations and future areas of work. The conclusion is missing a concise set of highlights or takeaways from this paper. Once again, as with the introduction, it should be emphasized what gap in literature this fills, and what the key contributions of this study are in comparison to the rest of the space.

o The limitations are the limited sample size and the confounding variable eye-to-tooth distance, which we now mention in the discussion. Key contributions are also mentioned in the end of the discussion.

---

## [Decision Letter · Decision Letter 1]

3 Aug 2023

PONE-D-22-35513R1The effect of 3D stereopsis and hand-tool alignment on learning effectiveness and skill transfer of a VR-based simulator for dental trainingPLOS ONE

Dear Dr. Kaluschke,

Thank you for submitting your manuscript to PLOS ONE. After careful consideration, we feel that it has merit but does not fully meet PLOS ONE’s publication criteria as it currently stands. Therefore, we invite you to submit a revised version of the manuscript that addresses the points raised during the review process.

We look forward to receiving your revised manuscript.

Kind regards,

Artak Heboyan, Ph.D.

Academic Editor

PLOS ONE

Journal Requirements:

Additional Editor Comments:

Please follow the comments of reviewers` and correct the manuscript accordingly.

Reviewers' comments:

Reviewer's Responses to Questions

**Comments to the Author**

1. If the authors have adequately addressed your comments raised in a previous round of review and you feel that this manuscript is now acceptable for publication, you may indicate that here to bypass the “Comments to the Author” section, enter your conflict of interest statement in the “Confidential to Editor” section, and submit your "Accept" recommendation.

Reviewer #1: All comments have been addressed

Reviewer #2: All comments have been addressed

2. Is the manuscript technically sound, and do the data support the conclusions?

Reviewer #1: Yes

Reviewer #2: Yes

3. Has the statistical analysis been performed appropriately and rigorously? 

Reviewer #1: Yes

Reviewer #2: Yes

4. Have the authors made all data underlying the findings in their manuscript fully available?

Reviewer #1: Yes

Reviewer #2: Yes

5. Is the manuscript presented in an intelligible fashion and written in standard English?

Reviewer #1: Yes

Reviewer #2: Yes

6. Review Comments to the Author

Reviewer #1: Dear authors,

Thank you for your revised manuscript.

I do feel that my comments have been addressed. Therefore I only have minor comments:

Paragraph 2. Related work:

In my humble opinion, some of the added information contain redundant information or justifications of your study which should be rephrased. For example:

Unnecessary justifications:

- "... In contrast, we use an HMD for both conditions and disable stereopsis in the monoscopic 3D condition" (more appropriate in the discussion if you want to add this information to the manuscript)

- "To produce 2D images, the simulator was engineered to output a single image to both eyes, which is essentially the same way we achieved it" (reading this sentence it sounds too emotional regarding the comments of the reviewer)

redundant information:

- "Their findings include that the participants from immersed groups either in preceptorship or training performed the anesthetic procedure faster and more accurately than those in the combined non-immersed groups. The authors conclude that the stereoscopic view from the HMD in immersed groups provided a better perception of depth when compared to the 2D monitor, makinginstrument navigation inside the mouth easier and leading to better performance results.

The results showed that participants without immersive displays had less accurate needle insertion points, though needle injection angle and depth were not significantly different between the groups. The needle insertion point here needs to found without haptic feedback. As such it differs considerably from the root canal opening, since the bur can touch the tooth with drilling disabled for orientation with the help of haptic feedback". Please decide, which paragraph you prefer to summarize this paper.

Reviewer #2: Comments have been addressed thoroughly. The paper is much clearer and a stronger work as a result of the authors revising the manuscript.

7. PLOS authors have the option to publish the peer review history of their article (what does this mean?). If published, this will include your full peer review and any attached files.

Reviewer #1: No

Reviewer #2: No

---

## [Author Response · Author response to Decision Letter 1]

19 Aug 2023

Dear reviewers,

Thank you for your constructive comments. In the following I will explain how we tried to address each of your brought up points:

Reviewer #1

• Paragraph 2. Related work:

In my humble opinion, some of the added information contain redundant information or justifications of your study which should be rephrased. For example:

Unnecessary justifications:

"... In contrast, we use an HMD for both conditions and disable stereopsis in the monoscopic 3D condition" (more appropriate in the discussion if you want to add this information to the manuscript)

o Thank you for pointing this out. We have removed the mentioned sentence all together, as we feel it didn’t fit well into the Discussion either.

• "To produce 2D images, the simulator was engineered to output a single image to both eyes, which is essentially the same way we achieved it" (reading this sentence it sounds too emotional regarding the comments of the reviewer)

o Thank you for pointing this out. We removed the emotional sounding phrase.

• redundant information:

"Their findings include that the participants from immersed groups either in preceptorship or training performed the anesthetic procedure faster and more accurately than those in the combined non-immersed groups. The authors conclude that the stereoscopic view from the HMD in immersed groups provided a better perception of depth when compared to the 2D monitor, makinginstrument navigation inside the mouth easier and leading to better performance results.

The results showed that participants without immersive displays had less accurate needle insertion points, though needle injection angle and depth were not significantly different between the groups. The needle insertion point here needs to found without haptic feedback. As such it differs considerably from the root canal opening, since the bur can touch the tooth with drilling disabled for orientation with the help of haptic feedback". Please decide, which paragraph you prefer to summarize this paper.

o Thank you for spotting this redundancy. We have removed the first paragraph.

Reviewer #2

• Comments have been addressed thoroughly. The paper is much clearer and a stronger work as a result of the authors revising the manuscript.

o Thank you very much.

---

## [Decision Letter · Decision Letter 2]

29 Aug 2023

The effect of 3D stereopsis and hand-tool alignment on learning effectiveness and skill transfer of a VR-based simulator for dental training

PONE-D-22-35513R2

Dear Dr. Kaluschke,

We’re pleased to inform you that your manuscript has been judged scientifically suitable for publication and will be formally accepted for publication once it meets all outstanding technical requirements.

Kind regards,

Artak Heboyan, Ph.D.

Academic Editor

PLOS ONE

Additional Editor Comments (optional):

Reviewers' comments:

Reviewer's Responses to Questions

**Comments to the Author**

1. If the authors have adequately addressed your comments raised in a previous round of review and you feel that this manuscript is now acceptable for publication, you may indicate that here to bypass the “Comments to the Author” section, enter your conflict of interest statement in the “Confidential to Editor” section, and submit your "Accept" recommendation.

Reviewer #1: All comments have been addressed

2. Is the manuscript technically sound, and do the data support the conclusions?

Reviewer #1: Yes

3. Has the statistical analysis been performed appropriately and rigorously? 

Reviewer #1: Yes

4. Have the authors made all data underlying the findings in their manuscript fully available?

Reviewer #1: Yes

5. Is the manuscript presented in an intelligible fashion and written in standard English?

Reviewer #1: Yes

6. Review Comments to the Author

Reviewer #1: Thank you very much for your great effort in addressing all the questions raised by the reviewers. I therefore have nothing more to add in this review process.

7. PLOS authors have the option to publish the peer review history of their article (what does this mean?). If published, this will include your full peer review and any attached files.

Reviewer #1: No

---

## [Editor Report · Acceptance letter]

25 Sep 2023

PONE-D-22-35513R2 

The effect of 3D stereopsis and hand-tool alignment on learning effectiveness and skill transfer of a VR-based simulator for dental training 

Dear Dr. Haddawy:

I'm pleased to inform you that your manuscript has been deemed suitable for publication in PLOS ONE. Congratulations! Your manuscript is now with our production department. 

Kind regards, 

on behalf of

Dr. Artak Heboyan 

Academic Editor

PLOS ONE